# Modeling organic aerosol over Central Europe: uncertainties linked to different chemical mechanisms, parameterizations, and boundary conditions

Lukáš Bartík<sup>1</sup>, Peter Huszár<sup>1</sup>, Jan Peiker<sup>1,2</sup>, Jan Karlický<sup>1</sup>, Ondřej Vlček<sup>2</sup>, and Petr Vodička<sup>3</sup>

Correspondence: Lukáš Bartík (lukas.bartik@matfyz.cuni.cz)

#### Abstract.

This study explores the uncertainties in modeling organic aerosol (OA) over Central Europe, focusing on the roles of chemical mechanisms, emission parameterizations, and boundary conditions. Organic aerosols, particularly secondary organic aerosols (SOAs), significantly influence climate, health, and visibility, comprising up to 90 % of submicron particulate matter. Using the Comprehensive Air Quality Model with Extensions (CAMx) coupled with the Weather Research and Forecast Model, sensitivity analyses were conducted to assess the impact of intermediate-volatility organic compounds (IVOCs), semi-volatile organic compounds (SVOCs), and chemical boundary conditions on primary and secondary organic aerosol concentrations.

Model evaluation against organic carbon measurements over the Czech Republic showed that including source-specific IVOC and SVOC emissions significantly improved CAMx's performance, particularly when using the 1.5-dimensional Volatility Basis Set framework with activated chemical aging. For example, the domain-averaged SOA concentrations increased by up to  $1.17~\mu g \, m^{-3}$  during summer when both IVOC and SVOC emissions were included. Furthermore, incorporating OA into the boundary conditions enhanced model predictions, with the accuracy of modeled organic carbon concentrations significantly improved during summer at some monitoring sites. Despite these improvements, challenges remain due to uncertainties in emission estimates, parameterization schemes, and the spatial resolution of the models.

The findings underscore the importance of refined parameterizations for IVOC and SVOC emissions, higher temporal and spatial resolution in chemical boundary conditions, and better representation of chemical aging. Addressing these gaps in future studies will further enhance the understanding and prediction of OA dynamics in regional air quality modeling.

## 1 Introduction

Atmospheric aerosols are liquid or solid particles suspended in the atmosphere, which have substantial climate impacts via direct and indirect radiative effects (Li et al., 2022; Arola et al., 2022), negatively affect human health (Arias-Pérez et al., 2020; Ain and Qamar, 2021), reduce visibility (Singh and Dey, 2012), and represent an undoubted environmental burden, especially

<sup>&</sup>lt;sup>1</sup>Department of Atmospheric Physics, Faculty of Mathematics and Physics, Charles University, Prague, V Holešovičkách 2, 18000 Prague 8, Czech Republic

<sup>&</sup>lt;sup>2</sup>Czech Hydrometeorological Institute, Na Šabatce 2050/17, 143 06 Prague 4, Czech Republic

<sup>&</sup>lt;sup>3</sup>Institute of Chemical Process Fundamentals of the Czech Academy of Science, Rozvojová 1/135, 165 02, Prague 6, Czech Republic

near large urban areas (Wu and Boor, 2021). Organic aerosol (OA) constitutes a significant fraction of the total aerosol in the submicron range (PM<sub>1</sub>; particles with an aerodynamic diameter less than 1  $\mu$ m). In terms of the total mass of PM<sub>1</sub>, OA can account for as much as 90 % (Jimenez et al., 2009; Zhang et al., 2011; Crippa et al., 2014; Chen et al., 2022). For instance, in Europe, Lanz et al. (2010) reported OA contributions to the total PM<sub>1</sub> mass ranging from 36 to 81 %, while Morgan et al. (2010) measured values between 20 and 50 %. Additionally, these contributions tend to be higher over urban areas compared to rural ones (Bressi et al., 2013; Sandrini et al., 2016). Recently, Chen et al. (2022) noted that oxygenated OA components, which serve as proxies for secondary organic aerosol (SOA), comprise between 43.7 and 100 % of the submicron OA mass in Europe. They also stated that solid fuel combustion-related OA components still represent a significant portion of the submicron OA mass, particularly during winter (on average 21.4 %). These experimental findings highlight the need to identify sources of OA and assess their contributions to total concentrations of OA. Only based on their knowledge is it possible to develop effective strategies for reducing overall OA concentrations. Chemical transport models (CTMs) are essential tools for achieving these goals.

Over the past two decades, the approaches used for OA modeling in CTMs have evolved considerably. Traditionally, the modeling of primary (directly emitted) organic aerosol (POA) in CTMs has assumed that POA is non-volatile and chemically inert. At the same time, SOA formation has typically been modeled using the gas-particle partitioning of condensable products originating from the oxidation of reactive organic gases (e.g., Strader et al., 1999; Schell et al., 2001; Byun and Schere, 2006). This partitioning has often been approximated by applying absorptive partitioning in a pseudo-ideal solution (Pankow, 1994; Odum et al., 1996). Contrary to the traditional assumptions about POA, the experimental results showed that POA is mostly semi-volatile under ambient conditions, and the gas-phase portion can undergo photochemical oxidation, resulting in SOA formation (e.g., Robinson et al., 2007; Donahue et al., 2009). These findings led Donahue and his colleagues to develop two unified frameworks for gas-particle partitioning and chemical aging of both POA and SOA, known as volatility basis sets (VBSs). The first of these frameworks, referred to as the 1-dimensional (1-D) VBS, describes the evolution of OA by employing a set of lumped semi-volatile OA species with their volatilities equally spaced in a logarithmic scale (the basis set) (Donahue et al., 2006). In terms of effective saturation concentrations ( $C^*$ ) at a thermodynamic temperature of 298 K, 1-D VBS typically range from  $10^{-2}$  to  $10^6$  µg m<sup>-3</sup>, which covers three subcategories of organic compounds: low volatility organic compounds (LVOCs;  $C^* = \{10^{-2}, 10^{-1}\} \text{ } \mu\text{g m}^{-3}$ ), semi-volatile organic compounds (SVOCs;  $C^* = \{10^0, 10^1, 10^2\}$  $\mu g m^{-3}$ ), and intermediate volatility organic compounds (IVOCs;  $C^* = \{10^3, 10^4, 10^5, 10^6\} \mu g m^{-3}$ ) (Donahue et al., 2009). The second framework, known as the 2-dimensional (2-D) VBS, describes the evolution of OA in a 2-D space defined by volatility and the degree of oxidation (Donahue et al., 2011, 2012). Later, Koo et al. (2014) developed a 1.5-dimensional (1.5-D) VBS approach that is based on the 1-D VBS framework but accounts for changes in the oxidation state of OA as well as its volatility using multiple reaction trajectories defined in the 2-D space of the 2-D VBS framework. However, despite these advances in OA modeling, CTMs still face challenges in accurately reproducing measured OA concentrations, mainly due to the underestimation of SOA concentrations.

Several studies have shown that using both the original 1.5-D VBS and its various modifications can significantly improve the model predictions of SOA in different regions of the Earth (e.g., Zhang et al., 2023; Jiang et al., 2021, 2019b; Yao et al., 2020;

Giani et al., 2019; Meroni et al., 2017; Ciarelli et al., 2017; Woody et al., 2016). These modifications typically involve changes to the number of basis sets, the physical parameters that characterize them, or both, and are often used in combination with emission estimates for missing SOA precursors. The term "missing SOA precursors" means IVOCs and SVOCs, as they are missing in traditional emission inventories used to create input emission data for CTMs. However, these studies also indicate that the degree of improvement achieved is accompanied by considerable uncertainty. This uncertainty is partly caused by the inaccuracy of a large number of parameters characterizing individual basic sets, such as, for example, effective enthalpies of vaporization of POA and SOA species, product mass yields for oxidation of IVOCs, and the reaction rates associated with these oxidations, which are constrained using data obtained from smog chamber experiments. It was demonstrated, for example, by Jiang et al. (2021), who significantly improved the modeled concentrations of OA and SOA over Europe during winter by using optimized parameters in the basis sets for POA and SOA originating from biomass burning. These optimized parameters were specifically determined to account for the losses of semi-volatile vapors on the walls of the smog chamber used in the experiments conducted to determine them. Similarly, Ciarelli et al. (2017) updated a modified 1.5-D VBS scheme with parameters determined based on novel smog chamber experiments focused on biomass burning and showed that these updates significantly improved the modeled concentrations of total OA and SOA over Europe. Also, Jiang et al. (2019b) created a modified 1.5-D VBS scheme, optimized using parameters based on current smog chamber experiments focused on emissions from diesel cars and biomass burning. They demonstrated that this modified VBS scheme improved the model performance for total OA as well as its components, including hydrocarbon-like OA, biomass-burning-like OA, and oxygenated OA components.

Another significant source of uncertainty in the modeled concentrations of OA and its components when using traditional emission inventories are the emission estimates of missing SOA precursors and the volatility distribution factors for POA emissions. Regarding IVOC emissions, many previous works (e.g., Meroni et al., 2017; Denier van der Gon et al., 2015; Koo et al., 2014; Tsimpidi et al., 2010) estimated them using a non-source-specific parameterization (IVOC =  $1.5 \times POA$ ) proposed by Robinson et al. (2007). However, this simplifying assumption has been partially addressed over time by establishing several source-specific parameterizations for IVOC emission estimates. These parameterizations were derived from smog chamber experiments conducted with emissions from various sources, such as biomass burning, diesel vehicles, and gasoline vehicles (e.g., Jiang et al., 2021; Giani et al., 2019; Ciarelli et al., 2017; Zhao et al., 2016, 2015; Jathar et al., 2014). Further, in order to account for missing SVOC emissions in model simulations employing VBS approaches, many researchers have opted to increase the amount of POA emissions by a factor of 3 (e.g., Jiang et al., 2021, 2019b; Li et al., 2020; Ciarelli et al., 2017, 2016; Matsui et al., 2014; Shrivastava et al., 2011; Hodzic et al., 2010; Tsimpidi et al., 2010). This adjustment is based on partitioning theory predictions aimed at compensating for missing gaseous emissions in the semi-volatile range (Ciarelli et al., 2017). As a result, the adjusted POA emissions are intended to represent the total primary organic matter spanning both the semi-volatile and lower-volatility ranges of the volatility spectrum, hereafter referred to as POM<sub>SV</sub>. Here, it is appropriate to note that this approach is in good agreement with the results of Denier van der Gon et al. (2015), who constructed a revised European bottom-up emission inventory for residential wood combustion accounting for SVOCs and showed that the revised emissions are higher than those in the previous inventory by a factor of 2–3 but with substantial inter-country variation. Recently, to move

beyond the theory-based approach to estimating POM<sub>SV</sub> mentioned above, Giani et al. (2019) relied on experimental data from Zhao et al. (2015) and Zhao et al. (2016), which examined emissions from diesel and gasoline vehicles, respectively. These studies provided not only estimates of IVOC emissions, scaled according to emissions of non-methane hydrocarbons, but also the complete volatility distributions of these emissions. Using these distributions, Giani et al. (2019) could determine the ratios between IVOC and POM<sub>SV</sub> emissions. Subsequently, using these ratios, they determined the POM<sub>SV</sub> emissions themselves.

In this study, we present two sensitivity analyses focused on various aspects related to the transport and chemistry of OA over Central Europe. The first analysis investigates how estimates of IVOC and POM<sub>SV</sub> emissions influence the modeled concentrations of OA, considering various model approaches to gas-phase chemistry and the chemical and thermodynamic processes associated with OA. In comparison with previous works devoted to a similar topic over Europe (e.g., Jiang et al., 2021; Giani et al., 2019; Ciarelli et al., 2017; Meroni et al., 2017; Ciarelli et al., 2016), which focused mainly on the winter period, we concentrate on the winter and summer periods in our study. The second sensitivity analysis evaluates the impact of large-scale transport of OA, i.e., the effect of incorporating OA data, including its partitioning into primary and secondary fractions, into chemical boundary conditions. Due to the size and location of the model domain we used, this type of sensitivity analysis is crucial.

## 2 Methodology

100

105

115

All model experiments used in this study were performed with the Comprehensive Air Quality Model with Extensions (CAMx) version 7.10 (Ramboll, 2020), which was offline coupled with the Weather Research and Forecast (WRF) Model version 4.2 (Skamarock et al., 2019) and the Model of Emissions of Gases and Aerosols from Nature (MEGAN) version 2.10 (Guenther et al., 2012). The two model experiments employed in both sensitivity analyses (henceforth referred to as CSwI and CVb) were taken from the work of Bartík et al. (2024), in which they represent the base simulations of the SOAP and VBS experiments. All additional experiments were conducted on the same Central European model domain as CSwI and CVb. This domain, characterized by a horizontal resolution of 9 km  $\times$  9 km, is centered over Prague (50.075° N, 14.440° E), Czech Republic (Fig. 1a). In addition, these additional experiments were carried out for the same period as CSwI and CVb (i.e., for the years 2018 and 2019) and used the same driving meteorological fields as these two experiments. A comprehensive description of the model domain and the configuration of the WRF model simulation used to produce the driving meteorological fields can be found in Bartík et al. (2024).

#### 2.1 CAMx and its configurations used

CAMx is a state-of-the-art Eulerian CTM designed to simulate all key processes involved in the transport and chemistry of pollutants. These processes include horizontal and vertical advection, horizontal and vertical diffusion, gas-phase and aerosol chemistry, and wet and dry deposition.

Gas-phase chemistry in the model experiments was simulated using two mechanisms that represent the essential chemical processes involved in the formation of ozone and SOA, including photolysis, oxidation by hydroxyl radicals, nitrate radicals,

Figure 1. Resolved model terrain altitude above sea level (in m) across the model domain used in the study (a). The locations of the measuring stations used for validation are shown in (b).

and ozone, and the formation and reactions of hydroperoxyl and organic peroxy radicals. The first was the fifth revision of the Carbon Bond mechanism version 6 (CB6r5), which includes 233 reactions among 87 species (62 state gases and 25 radicals) (Ramboll, 2020). CB6r5 is a lumped-structure mechanism that represents groups of volatile organic compounds (VOCs) using surrogate species based on chemical structure and bond types, including aliphatic, olefinic, and aromatic bonds. In addition to these lumped surrogates, the mechanism also treats several compounds explicitly, such as isoprene, formaldehyde, acetaldehyde, ethanol, glyoxal, methylglyoxal, and glycolaldehyde. The second mechanism was SAPRC07TC, the 2007 version of the

30 Statewide Air Pollution Research Center mechanism (Carter, 2010; Hutzell et al., 2012), which includes 565 reactions among 117 species (72 state gases and 45 radicals) in its CAMx implementation (Ramboll, 2020). While it also applies lumping, primarily based on VOC reactivity, it retains a much larger number of VOCs and their oxidation products explicitly. Both mechanisms were solved numerically using the Euler Backward Iterative method developed by Hertel et al. (1993).

These two mechanisms were selected to evaluate how differences in gas-phase chemistry formulations affect SOA production while keeping the SOA treatment unchanged. Their use also reflects practical constraints in the available configurations of the CAMx model: when setting up the experiments described in Sect. 2.4, we considered only those combinations of gas-phase mechanisms and OA modules that are directly supported, as other pairings would have required additional modifications to the model code, which we sought to avoid.

Further, we selected the coarse/fine (CF) aerosol scheme (Ramboll, 2020) to couple aerosol processes with gas-phase chemistry, as it is the only option in CAMx that supports all configurations of gas-phase mechanisms and organic aerosol modules adopted in this study (see Sect. 2.4). This scheme divides the aerosol size distribution into two static, non-interacting modes (fine and coarse), within which aerosols are treated as internally mixed and monodisperse in size. Primary aerosol species can be represented in one or both modes, while all secondary aerosol species are modeled exclusively in the fine mode. Coarsemode aerosol species are treated as non-volatile, chemically inert, and subject only to emission, transport, and removal by dry and wet deposition. In contrast, while all fine-mode aerosol species undergo the same physical processes as coarse-mode species, many of them can also participate in gas-particle partitioning, which is calculated based on the thermodynamic equilibrium assumption and applied separately to inorganic and organic aerosol species. Specifically, to predict the composition and physical state of inorganic aerosol species, we used the thermodynamic equilibrium model ISORROPIA version 1.7 (Nenes et al., 1998, 1999), which models the sodium-ammonium-chloride-sulfate-nitrate-water aerosol system, including the mutual deliquescence behavior of multicomponent salt particles. One of two modules can be used to control organic gas-particle partitioning and oxidation chemistry in CAMx version 7.10 (Ramboll, 2020). The first is the Secondary Organic Aerosol Processor (SOAP) version 2.2, originally developed by Strader et al. (1999). The second is the 1.5-D VBS module developed by Koo et al. (2014). Since both were used in our model experiments (see Sect. 2.4), we briefly describe them and highlight their differences in the next subsection. The CF scheme also accounts for aqueous aerosol formation in resolved cloud water, using a modified version of the RADM (Regional Acid Deposition Model) aqueous chemistry algorithm (Ramboll, 2020), originally developed by Chang et al. (1987). This algorithm includes, among other processes, the aqueous-phase formation of SOA from water-soluble precursors such as glyoxal, methylglyoxal, and glycolaldehyde (Ortiz-Montalvo et al., 2012; Lim et al., 2013).

## **2.2 SOAP and 1.5-D VBS**

respectively.




Here, we briefly describe both modules that govern the OA chemistry in CAMx version 7.10. More information can be found in Ramboll (2020).

Additionally, we used the CAMx wet deposition model (Ramboll, 2020) to solve the wet deposition of gases and aerosols. To calculate the dry deposition of gases and aerosols, we used the methods of Zhang et al. (2003) and Zhang et al. (2001),

SOAP version 2.2, hereafter referred to simply as SOAP, treats POA using the traditional assumptions mentioned in the introduction; thus, it considers POA to be a single non-volatile species that does not evolve chemically. It also accounts for the oxidation of four anthropogenic gaseous species (benzene, toluene, xylene, and anthropogenic IVOCs) producing two condensable gas (CG) species (a more-volatile compound with a saturation concentration,  $C^{\circ}$ , of 14  $\mu$ g m<sup>-3</sup> and a less-volatile compound with  $C^{\circ} = 0.31 \ \mu$ g m<sup>-3</sup>), as well as one non-volatile aerosol-phase species. Similarly, the oxidation of three biogenic volatile organic compounds (BVOCs) (isoprene, monoterpenes, and sesquiterpenes) generates two distinct CG species (more-volatile with  $C^{\circ} = 26 \ \mu$ g m<sup>-3</sup> and less-volatile with  $C^{\circ} = 0.45 \ \mu$ g m<sup>-3</sup>) and one non-volatile species. All saturation concentrations ( $C^{\circ}$ ) are reported at 300 K. The more-volatile and less-volatile CG species from both anthropogenic and biogenic precursors are redistributed between the gas and aerosol phases following the pseudo-ideal solution theory (Strader et al., 1999).

The 1.5-D VBS scheme uses five basis sets to represent different degrees of oxidation in ambient OA: three basis sets for freshly emitted OA (originating from meat cooking, biomass burning, and other anthropogenic sources) and two basis sets for chemically aged, oxygenated OA (including both anthropogenic and biogenic sources). Each basis set comprises five volatility bins with  $C^{\circ} = \{10^{-1}, 10^{0}, 10^{1}, 10^{2}, 10^{3}\} \, \mu \mathrm{g m}^{-3}$  at 298 K. Each volatility bin includes two surrogate species that represent the particle and gas phases of a single lumped organic species.





For clarity, it is important to note that although the properties of the surrogate species in the lowest volatility bin were estimated assuming  $C^{\circ} = 10^{-1} \ \mu \mathrm{g m^{-3}}$ , they in fact represent all OA of a given type with  $C^{\circ} \leq 10^{-1} \ \mu \mathrm{g m^{-3}}$  and are treated as non-volatile. In practice, this means that whenever the gas-phase surrogate species in the lowest volatility bin is produced in any of the basis sets via chemical aging, it is assumed to immediately condense into its corresponding particle-phase surrogate species, which is treated as non-volatile and does not evaporate. For lumped species representing volatility bins other than the lowest bin in each basis set, partitioning between the gas and particle phases is again calculated using the pseudo-ideal solution theory (Strader et al., 1999), in the same way as in SOAP.

The chemical aging of OA is modeled by redistributing OA mass along predefined pathways within and between the basis sets, decreasing its volatility while simultaneously increasing its oxidation state. The gas-phase hydroxyl radical reaction rates for the chemical aging of POA and anthropogenic SOA, excluding those from biomass burning, are assumed to be  $4 \times 10^{-11}$  and  $2 \times 10^{-11}$  cm<sup>3</sup> molecule<sup>-1</sup> s<sup>-1</sup>, respectively. In the default configuration of the 1.5-D VBS scheme, the chemical aging of biogenic SOA and SOA originating from biomass burning (both anthropogenic and biogenic) is disabled. This configuration was used in the CVb experiment conducted in this study (see Sect. 2.4.1). The CVa experiment, also carried out in this study, retained the aging treatments included in CVb and, in addition, activated the aging of biogenic and biomass-burning SOA, applying the same reaction rate of  $2 \times 10^{-11}$  cm<sup>3</sup> molecule<sup>-1</sup> s<sup>-1</sup> as for anthropogenic SOA.

The 1.5-D VBS scheme incorporates the oxidation of traditional anthropogenic and biogenic gaseous precursors of SOA used in SOAP, including benzene, toluene, xylene, isoprene, monoterpenes, and sesquiterpenes. Similar to SOAP, it also includes the oxidation of IVOCs. However, unlike SOAP, which represents all anthropogenic IVOC emissions with a single surrogate species, the 1.5-D VBS scheme employs four source-specific surrogate species for IVOC emissions, corresponding to gasoline vehicles, diesel vehicles, other anthropogenic sources, and biomass burning.

Also, unlike SOAP, which maps POA emissions from all anthropogenic sources to a single non-volatile aerosol species, the 1.5-D VBS scheme allocates POA emissions based on their source category to one of the three basis sets representing freshly emitted OA. Within the assigned basis set, the emissions are further redistributed across all volatility bins using source-specific volatility distribution factors. The scheme distinguishes between POA emissions from gasoline vehicles, diesel vehicles, meat cooking, other anthropogenic sources, and biomass burning, applying a separate set of volatility distribution factors to each of these source categories.

## 2.3 Input emission data and chemical boundary conditions








The input emission data employed in the model experiments can be categorized into two groups. The first group comprises biogenic and traditional anthropogenic emissions, which remain consistent across all the model experiments except for POA emissions, as will be discussed in more detail in Sect. 2.4.1. A more comprehensive description of these emissions and their preparation for the model experiments can be found in Bartík et al. (2024). In summary, anthropogenic emissions outside the territory of the Czech Republic were sourced from the CAMS (Copernicus Atmosphere Monitoring Service) European anthropogenic emissions - Air Pollutants inventory version 4.2 (CAMS-REG-v4.2; Kuenen et al., 2021) for the year 2018, with a spatial resolution of approximately  $0.05^{\circ} \times 0.1^{\circ}$ . Within the Czech Republic, we used high-resolution emissions from the Register of Emissions and Air Pollution Sources (REZZO – Registr emisí a zdrojů znečištění ovzduší) for the year 2018. along with emissions from the ATEM Traffic Emissions dataset for the year 2016. The REZZO emissions were provided by the Czech Hydrometeorological Institute (https://www.chmi.cz), while the ATEM dataset was supplied by ATEM (Ateliér ekologických modelů - Studio of Ecological Models; https://www.atem.cz). The raw anthropogenic emissions were interpolated into the model grid, and the temporal disaggregation of annual emission totals into hourly fluxes and the speciation of nonmethane volatile organic compounds (NMVOCs) and fine particulates into species considered by the CAMx mechanisms (both gas-phase and aerosol) were carried out using the FUME emission preprocessor (Benešová et al., 2018; Belda et al., 2024), incorporating the temporal and speciation profiles provided by Denier van der Gon et al. (2011) and Passant (2002), respectively. BVOC emissions were calculated using MEGAN version 2.1 (Guenther et al., 2012) based on the hourly WRF output fields.

The second group includes IVOC and SVOC emissions from anthropogenic sources. As these emissions are not part of the used emission inventories, we estimated them using sector-specific or sector-non-specific parameterizations, which will also be discussed in greater detail in Sect. 2.4.1.

To force the model experiments at the boundary of the model domain, we employed two distinct sets of chemical boundary conditions (CBCs). The first set consists of time-space invariant concentrations of ozone and its precursors, including several reactive nitrogen compounds and NMVOCs, as outlined in Table S1 in the Supplement. Their values reflect typical background concentrations over Europe, derived from simulations performed by Huszar et al. (2020b) over a large European domain with a horizontal resolution of 27 km. Hereafter, we refer to this set as the default CBCs. As indicated in Table S1, the number of chemical species used for the default CBCs varies slightly depending on the chosen gas-phase mechanism, which reflects differences in their formulations.

**Table 1.** Model setup (gas-phase chemistry mechanism, OA chemistry module, and additional aging) and the inclusion of IVOC and SVOC emission estimates in the individual experiments of the first sensitivity analysis. Notes: <sup>a</sup> The additional aging refers to the aging of SOA originating from biomass burning and biogenic sources, which is disabled by default in the 1.5-D VBS scheme.

| Experiment | Gas-phase chemistry mechanism | OA chemistry module | Addiotional aging <sup>a</sup> | IVOC emission estimates included | SVOC emission estimates included |
|------------|-------------------------------|---------------------|--------------------------------|----------------------------------|----------------------------------|
| CSnI       | CB6r5                         | SOAP                | No                             | No                               | No                               |
| CSwI       | CB6r5                         | SOAP                | No                             | Yes                              | No                               |
| SSnI       | SAPRC07TC                     | SOAP                | No                             | No                               | No                               |
| SSwI       | SAPRC07TC                     | SOAP                | No                             | Yes                              | No                               |
| CVb        | CB6r5                         | 1.5-D VBS           | No                             | Yes                              | Yes                              |
| CVa        | CB6r5                         | 1.5-D VBS           | Yes                            | Yes                              | Yes                              |

The second set of CBCs, hereafter referred to as the EAC4 CBCs, was developed using the monthly averaged fields of the CAMS global reanalysis (EAC4) dataset (Inness et al., 2019). Specifically, we used mean monthly concentrations of all gas-phase and aerosol species provided in the EAC4 dataset, as listed in Table S2 of the Supplement. Compared to the default CBCs, the EAC4 CBCs contain several gas-phase species absent from the default set, such as methane, ethane, propane, and sulfur dioxide. In contrast, several species present in the default CBCs are absent from the EAC4 CBCs, including nitrous acid, methanol, ethanol, formaldehyde, acetaldehyde, toluene, and xylene. More importantly, the EAC4 CBCs differ fundamentally from the default CBCs by incorporating aerosol species, namely sea salt, dust, sulfate, hydrophobic black carbon, hydrophobic organic matter, and hydrophilic organic matter. The mapping used to convert these aerosol species to those recognized by the CF scheme is also provided in Table S2. The inclusion of both hydrophobic and hydrophilic organic matter in the EAC4 dataset is particularly important for the second sensitivity analysis. The mapping of these two aerosol species to the OA species recognized by the CF scheme is described in Sect. 2.4.2.

#### 2.4 Sensitivity analyses



## 2.4.1 First sensitivity analysis – impact of OA module and IVOC/POM<sub>SV</sub> emissions

As mentioned in the Introduction, the aim of the first sensitivity analysis is to examine how estimates of IVOC and POM<sub>SV</sub> emissions influence the modeled concentrations of OA, considering various model approaches to gas-phase chemistry and the chemical and thermodynamic processes associated with OA. To achieve the objective of this sensitivity analysis, we used CSwI and CVb and developed four additional experiments, which include CSnI, SSnI, SSwI, and CVa (Table 1). These experiments can be split into two groups based on the module used to model OA chemistry.

The first group comprises CSnI, CSwI, SSnI, and SSwI, all utilizing SOAP. At the same time, CB6r5 was applied to model gas-phase chemistry in CSnI and CSwI, while SAPRC07TC was used for this purpose in SSnI and SSwI. As for IVOC emissions, both CSwI and SSwI considered them, whereas CSnI and SSnI were conducted without their inclusion. The second

group includes CVb and CVa, in which OA chemistry was modeled using the 1.5-D VBS scheme, the gas-phase chemistry was modeled using CB6r5, and identical emission estimates of both IVOC and POM<sub>SV</sub> were employed. In both experiments, the chemical aging of POA and anthropogenic SOA (excluding biomass burning sources) was included, using gas-phase hydroxyl radical reaction rates of  $4 \times 10^{-11}$  and  $2 \times 10^{-11}$  cm<sup>3</sup> molecule<sup>-1</sup> s<sup>-1</sup>, respectively (see Sect. 2.2). The only difference between CVb and CVa lies in the treatment of chemical aging for SOA originating from biogenic emissions and biomass burning. While CVb used the default 1.5-D VBS configuration with aging of these SOA types disabled, CVa enabled their aging using the same reaction rate as for anthropogenic SOA.

Regarding the IVOC and POM<sub>SV</sub> emission estimates used in individual experiments, Table 2 summarizes all the parameterizations applied to calculate them for the respective anthropogenic sources. It is also important to note that in all four experiments utilizing SOAP, we applied POA emissions obtained from the traditional emission databases mentioned in Sect. 2.3. Given that these POA emissions come from traditional emission inventories, we assumed that they do not account for missing SVOCs, consistent with the approach adopted in many previous studies cited in the Introduction. However, as we mentioned in Sect. 2.2, POA emissions in the 1.5-D VBS scheme are redistributed across all the volatility bins within the appropriate basis set, based on their source, and should therefore include the missing SVOCs. Consequently, in the two experiments employing the 1.5-D VBS scheme, we substituted the original POA emission estimates with those for POM<sub>SV</sub> to ensure inclusion of the missing SVOCs. Apart from accounting for the missing SVOCs, POM<sub>SV</sub> is otherwise treated identically to POA within the 1.5-D VBS scheme.

As can be seen in Table 2, we employed the IVOC emission estimates in SSwI as those used by Bartík et al. (2024) in CSwI. Furthermore, in CVa, we applied the identical IVOC and  $POM_{SV}$  emission estimates used by Bartík et al. (2024) in CVb.  $POM_{SV}$  emissions from gasoline and diesel vehicles were estimated using the approach proposed by Giani et al. (2019). For residential biomass burning and other anthropogenic sources, we followed the approach of Jiang et al. (2021) and adopted theoretically derived values ( $POM_{SV} = 3 \times POA$ ). Both of these approaches have been described in the Introduction. Following the method of Giani et al. (2019), we allocated  $POM_{SV}$  emissions to the volatility basis set using the volatility distribution factors proposed by Zhao et al. (2016) for gasoline vehicles and by Zhao et al. (2015) for diesel vehicles. Similarly,  $POM_{SV}$  emissions from residential biomass burning and other anthropogenic sources were allocated using the factors proposed by May et al. (2013) and Robinson et al. (2007), respectively, which serve as the default factors for these sources in CAMx version 7.10 (Ramboll, 2020). All of these allocation factors are provided in Table S3 in the Supplement. Finally, all experiments in this sensitivity analysis were conducted using the default chemical boundary conditions and did not account for aerosols outside the model domain.

## 2.4.2 Second sensitivity analysis – impact of OA composition in chemical boundary conditions

In order to study how the inclusion of OA in the chemical boundary conditions affects its concentrations inside the model domain, we established CSwI and CVb as two reference experiments since no aerosols were included in their CBCs. Following this, we conducted three sensitivity experiments for each of the reference experiments, namely Sp0s100, Sp50s50, and Sp100s0 for CSwI and Vp0s100, Vp50s50, and Vp100s0 for CVb. Each of these sensitivity experiments was performed using the same

**Table 2.** Parameterizations of the IVOC and POM<sub>SV</sub> emission estimates used in the individual experiments of the first sensitivity analysis. The individual parameterizations were taken from <sup>a</sup> Giani et al. (2019), <sup>b</sup> Jiang et al. (2021), <sup>c</sup> Robinson et al. (2007).

| Experiment | Parameterization for | Gasoline vehicles (GV)                                | Diesel vehicles (DV)                                  | Biomass<br>burning (BB)    | Other sources (OS)        |
|------------|----------------------|-------------------------------------------------------|-------------------------------------------------------|----------------------------|---------------------------|
| CSnI       | IVOC                 | 0                                                     | 0                                                     | 0                          | 0                         |
| CSwI       | IVOC                 | $0.0397 \times \text{NMVOC}_{\text{GV}}{}^{a}$        | $1.2748 \times \text{NMVOC}_{\text{DV}}{}^{\text{a}}$ | $4.5\times POA_{BB}{}^{b}$ | $1.5 \times POA_{OS}^{c}$ |
| SSnI       | IVOC                 | 0                                                     | 0                                                     | 0                          | 0                         |
| SSwI       | IVOC                 | $0.0397 \times \text{NMVOC}_{\text{GV}}{}^{a}$        | $1.2748 \times \text{NMVOC}_{\text{DV}}{}^{\text{a}}$ | $4.5\times POA_{BB}{}^{b}$ | $1.5 \times POA_{OS}^{c}$ |
| CVb        | IVOC                 | $0.0397 \times \text{NMVOC}_{\text{GV}}{}^{a}$        | $1.2748 \times \text{NMVOC}_{\text{DV}}{}^{\text{a}}$ | $4.5\times POA_{BB}{}^{b}$ | $1.5 \times POA_{os}^{c}$ |
|            | $POM_{SV}$           | $IVOC_{GV}$ / $4.62^a$                                | $IVOC_{DV}$ / $2.54^a$                                | $3\times POA_{BB}{}^{b}$   | $3\times POA_{OS}{}^{b}$  |
| CVa        | IVOC                 | $0.0397 \times \text{NMVOC}_{\text{GV}}{}^{\text{a}}$ | $1.2748 \times \text{NMVOC}_{\text{DV}}{}^{\text{a}}$ | $4.5\times POA_{BB}{}^{b}$ | $1.5 \times POA_{OS}^{c}$ |
|            | $POM_{SV}$           | $IVOC_{GV}$ / $4.62^a$                                | $IVOC_{DV}$ / $2.54^a$                                | $3\times POA_{BB}{}^b$     | $3\times POA_{OS}{}^{b}$  |

model setup and IVOC and  $POM_{SV}$  parameterizations as its corresponding reference experiment, but with CBCs that differed from those prescribed in the reference experiments (i.e., the default CBCs) in both gas-phase and aerosol species. Specifically, in these sensitivity experiments, we used three modifications of the EAC4 CBCs, each differing in the proportions of POA and SOA within the total OA (Table 3).





We opted for this approach due to the uncertainties involved in mapping OA from the EAC4 dataset to the OA species recognized by the CF scheme. This mapping was necessary because while the EAC4 dataset provides OA concentrations categorized into hydrophobic and hydrophilic components, the CF scheme uses POA and SOA surrogate species (Table S2 in the Supplement). However, since the proportions of POA and SOA in both hydrophobic and hydrophilic OAs were unknown, we decided to merge both species into the total OA. Next, we considered three scenarios for redistributing the total OA between POA and SOA. The first two scenarios represented extreme cases: in the first scenario, we treated the total OA as entirely composed of POA, while in the second scenario, we treated it as consisting wholly of SOA. The third scenario assumed a 50 percent share of both POA and SOA. Subsequently, we used these scenarios to obtain three pairs of boundary conditions for POA and SOA. We then added the same EAC4-derived boundary conditions for all gas-phase species and for the remaining remapped aerosol species to each pair of these boundary conditions, yielding the three modifications of the EAC4 CBCs mentioned above. As we have indicated in Table 3, the modification prepared using the first scenario was used in Sp100s0 and Vp100s0, the modification prepared using the second scenario was used in Sp0s100 and Vp0s100, and the modification prepared using the third scenario was used in Sp50s50 and Vp50s50.

Another challenge we encountered was determining how to redistribute POA and SOA within the EAC4 CBC modifications to their respective surrogate species used in both SOAP and the 1.5-D VBS scheme. To address this issue, we established two simplifying assumptions regarding the distribution of surrogate species at the boundary of the model domain. First, we assumed that the relative contributions of the surrogate species to total POA and to total SOA are spatially invariant but vary seasonally.

**Table 3.** Percentage share of POA and SOA in the total OA at the boundary of the model domain in the individual experiments of the second sensitivity analysis.

| Experiment | POA   | SOA   |  |
|------------|-------|-------|--|
| CSwI       | 0 %   | 0 %   |  |
| Sp0s100    | 0%    | 100 % |  |
| Sp50s50    | 50 %  | 50 %  |  |
| Sp100s0    | 100 % | 0%    |  |
| CVb        | 0%    | 0%    |  |
| Vp0s100    | 0%    | 100 % |  |
| Vp50s50    | 50 %  | 50 %  |  |
| Vp100s0    | 100 % | 0 %   |  |

Second, for each season, we calculated the domain-averaged mean concentrations of the surrogate SOA species separately for each reference experiment, and derived the seasonal fractions by dividing the concentration of each surrogate by the total SOA concentration in that experiment and season, defined as the sum of all SOA surrogate species (see Sect. 2.2). We applied the same procedure for the surrogate POA species. These seasonal fractions, used in their normalized form, were then applied as factors to allocate the total POA and SOA concentrations at the boundary to their respective surrogate species within all three modifications of boundary conditions. Table S4 in the Supplement shows the factors used to redistribute SOA to the surrogate species in SOAP. Tables S5 and S6 in the Supplement provide the factors used to redistribute SOA and POA to their surrogate species in the 1.5-D VBS scheme.

#### 2.5 Validation





We conducted a detailed validation of both CSwI and CVb in our previous study (Bartík et al., 2024). This validation included an assessment of modeled predictions against measurements for fine particulate matter (PM<sub>2.5</sub>) and its components (ammonium, nitrates, sulfates, elemental carbon, and organic carbon), nitrogen dioxide, and sulfur dioxide. Consequently, our focus here was solely on evaluating the modeled OA concentrations within the individual model experiments of both sensitivity studies. For this evaluation, we used organic carbon (OC) measurements collected at stations in the Czech Republic (Table S7 in the Supplement), which covered at least part of the modeled period of 2018 and 2019. Some of these measurements were obtained during two specific measuring campaigns, while the remainder were collected at the Prague–Suchdol and Košetice stations.

The first campaign was conducted at the Kosmos, Ropice, and Vrchy stations in the northeastern part of the Czech Republic, specifically in the Třinecko area (Seibert et al., 2020). The second campaign took place at the Švermov, Libušín, and Zbečno stations, located in Central Bohemia, specifically within the Kladensko area (Seibert et al., 2021). The locations of all these stations are shown in Fig. 1b. Both campaigns were divided into winter and summer phases, each lasting approximately one

month; the specific schedules can be found in Table S7 in the Supplement. During both campaigns, individual OC samples were continuously collected over 12-hour periods using sampling streams and collection heads to ensure representative sampling of the  $PM_{2.5}$  fraction. Subsequently, the mean 12-hour OC concentrations were determined from the collected samples. For validation purposes, we further derived mean daily OC concentrations by averaging the mean 12-hour concentrations; the rationale for this approach is explained in the final paragraph of this section. At the same time, air temperature, wind speed, and relative humidity were measured at the sampling stations, enabling the calculation of their mean daily values and the validation of these meteorological variables as well. The data from both campaigns were provided by the Czech Hydrometeorological Institute (https://www.chmi.cz).

The collection of  $PM_{10}$  (particulate matter with a diameter  $\leq 10~\mu m$ ) samples at the Prague–Suchdol station took place every fourth day for 24 h, from 2 January 2018, 09:00 UTC to 31 May 2018, 08:00 UTC. Starting on 27 March 2018, the sampling began at 08:00 UTC instead of 09:00 UTC. The samples were collected on prebaked (3 h, 800 °C) quartz fiber filters (Tissuequartz, Pall, 47 mm) using a Leckel sampler (Leckel GmbH, Germany). The filter cuts were analyzed for OC concentrations by a thermal-optical carbon analyzer (Sunset Laboratory Inc., USA) using the EUSAAR2 protocol (Cavalli et al., 2010). The resulting mean daily OC concentrations were corrected to blank.

The measurements of OC at the Košetice station in the Vysočina Region used for validation were obtained from the EBAS database (https://ebas-data.nilu.no/default.aspx) and cover the period from 1 January 2018, 02:00 UTC, to 31 December 2019, 02:00 UTC. These measurements represent OC within the  $PM_{2.5}$  fraction and were collected at 4-hour intervals, with each sample covering a 4-hour period. We derived mean daily OC concentrations from these data as 24-hour averages that follow the sampling schedule at the station (i.e., from 02:00 UTC on a given day to 02:00 UTC on the following day).

In order to validate the modeled values of air temperature, wind speed, and relative humidity at the Košetice and Prague–Suchdol stations, we used observational data provided by the Czech Hydrometeorological Institute from professional meteorological stations. Since the Košetice station is a professional meteorological station, we used the data directly from this site. Although the Prague–Suchdol station is an air quality station with accompanying meteorological measurements, the relevant meteorological data from it were not included among the data provided. Therefore, we used the data from the Prague–Kbely station, which was selected as a representative professional station for the Prague–Suchdol site.

The modeled mean daily OC concentrations were compared with the corresponding measured concentrations using several statistical measures, including mean bias (MB), root mean square error (RMSE), normalized mean square error (NMSE), the index of agreement (IOA), and the fraction of predictions within a factor of two of observations (FAC2). The definitions of these measures are provided in Eqs. (S1)–(S5) in the Supplement. For each of the stations, we considered only the modeled mean daily OC concentrations that corresponded to the days with the available OC measurements. Because the sampling periods varied among the stations used for validation (with the longest being 24 h at the Prague–Suchdol station), we opted to compare daily OC concentrations to ensure at least consistency in the duration of sampling periods across all the stations, even though the start and end times of these periods differ, as noted above. Since CAMx was configured to output hourly averaged concentrations, we were able to construct the modeled mean daily OC concentrations that exactly matched the sampling periods at each of the stations. To evaluate the modeled and observed values of mean daily air temperature, wind speed, and relative

humidity, we employed the MB, RMSE, NMSE, and IOA. For this meteorological assessment, we likewise used only the mean daily modeled and observed values of the meteorological variables from the days when the OC measurements were available, with each daily value constructed to follow the start and end times of the 24-hour sampling periods at the individual stations.

## 3 Results and discussion




### 3.1 Validation of meteorological variables

Figure S1 in the Supplement compares the observed and modeled mean daily temperatures, wind speeds, and relative humidities at the Prague–Kbely and Košetice stations across the individual seasons. Figure S2 in the Supplement provides similar comparisons at the stations used in both campaigns during their winter and summer phases. Our focus here is on the differences between the modeled and observed daily means of these meteorological variables. Figure 2 depicts these differences at the Prague–Kbely and Košetice stations throughout the respective seasons. Figure 3 illustrates these differences at the stations involved in both campaigns during their winter and summer phases. Additionally, Table S8 in the Supplement summarizes the statistical comparison of the modeled and observed daily means of these meteorological variables at all the stations.

The WRF model generally tends to slightly underestimate the mean daily temperatures, typically up to 1.5 K (Figs. 2a, b and 3a–d). An exception to this trend is observed at the Prague–Kbely station, where the model slightly overestimates the mean daily temperatures (Fig. 2b). The maximum differences in the mean daily temperatures only exceptionally exceeded 4 K (Fig. 3d). The NMSEs for the mean daily temperatures reached the lowest values (mainly below  $5 \times 10^{-3}$  %) among all the NMSEs evaluated. At the same time, the IOAs for the mean daily temperatures achieved the highest values (mostly above 0.9) among all the IOAs assessed.

**Figure 2.** Differences between the modeled and observed mean daily values of air temperature (in K) (**a**, **b**), wind speed (in m s<sup>-1</sup>) (**c**, **d**), and relative humidity (in %) (**e**, **f**) at the Košetice station (**a**, **c**, **e**) during the winter (DJF), spring (MAM), summer (JJA), and autumn (SON) of 2018 and 2019, and at the Prague–Kbely station (**b**, **d**, **f**) during January and February (JF) 2018 and the spring of 2018. Blue dotted lines indicate the level of zero difference.

**Figure 3.** Differences between the modeled and observed mean daily values of air temperature (in K) (**a–d**), wind speed (in m s<sup>-1</sup>) (**e–h**), and relative humidity (in %) (**i–l**) at the Kosmos, Ropice, and Vrchy stations in the Třinecko area during the winter (**a**, **e**, **i**) and summer (**c**, **g**, **k**) phase of the campaign, and at the Švermov, Libušín, and Zbečno stations in the Kladensko area during the winter (**b**, **f**, **j**) and summer (**d**, **h**, **l**) phase of the campaign. Blue dotted lines indicate the level of zero difference.

Regarding the mean daily wind speeds, the model has the tendency to overestimate them (Figs.2c, d and 3e–h), except for the Prague–Kbely station (Fig. 2d). At the Prague–Kbely and Košetice stations, the model showed a reasonable accuracy of their predictions in all the seasons, with IOAs exceeding 0.85 and NMSEs mostly below 10 %. In contrast, at the stations involved in both campaigns, the model overestimated them during both phases (MB = 3.2– $4.3 \text{ m s}^{-1}$  in the winter phases and MB = 1.1– $2.2 \text{ m s}^{-1}$  in the summer phases). At the same time, IOAs ranged from 0.1 to 0.54 and NMSEs between 29 % and 1320 %.




Figures 2e, f and 3i–1 demonstrate that the model somewhat underestimates the mean daily relative humidities at most of the stations, usually up to 9.5 %. At the same time, the maximum differences in the daily relative humidities only sporadically exceeded 20 %. The IOAs for the mean daily relative humidities exceeded 0.7 at most of the stations, while the NMSEs ranged between 0.5 % and 3.6%.

The overestimation of wind speed by the WRF model over the Central European domain with the same or similar horizontal resolution that we employed, especially in the winter months, was also pointed out in several papers (Huszar et al., 2020a; Karlický et al., 2020; Liaskoni et al., 2023; Bartík et al., 2024). This overestimation could be one of the sources of underestimating OA concentrations in all the experiments analyzed here. Aksoyoglu et al. (2011) showed that the reduced modeled wind speeds during observed periods of low wind can increase PM<sub>2.5</sub> concentrations by a factor of 2–3. Since OA usually forms a significant part of PM<sub>2.5</sub>, it can be assumed that similar increases also occur in OA concentrations under the mentioned conditions.

To further clarify the possible causes of the more substantial model overestimation of the mean daily wind speeds at the stations in the Kladensko area (Figs. 3f, h and S2f, h), we compared both the modeled and observed values of these wind speeds at the three stations with the corresponding wind speeds measured at four professional meteorological stations in Prague, as shown in Fig. S3 in the Supplement. As can be seen, the modeled wind speeds at the stations in the Kladensko area are more accurately represented by the Prague stations, located approximately 15–45 km away, than by the local stations themselves. This finding suggests that the campaign's stations were placed in locations where the wind field was more substantially influenced by nearby obstacles, such as buildings and trees, and/or by the contours of the surrounding terrain. These same factors likely contributed to the more notable model overestimation of the mean daily wind speeds at some of the stations in the Třinecko area (Figs. 3e, g and S2e, g).

Finally, it is also worth noting that the model biases in the other studied meteorological conditions may influence the modeled aerosol concentrations. For example, the lower modeled temperatures lead to an underestimation of gas-phase reaction rates due to their temperature dependence, but they may also enhance gas-to-particle partitioning. The negative bias in the modeled relative humidity compared to the observations affects particle size and density, as both are influenced by the aerosol water content determined by the local humidity. This implies that the model underestimates the aerosol water content, resulting in smaller particles, which may, in turn, slow down removal by deposition processes.

## 3.2 Sensitivity on OA module and IVOC/POM<sub>SV</sub> emissions






In this subsection, we present and discuss the spatial distributions of the mean seasonal impacts on the near-surface concentrations of POA and SOA (i.e., concentrations in the first model layer, which spanned approximately 50 m in vertical extent) in the
individual experiments of the first sensitivity analysis. Specifically, we focus on these impacts during the winters and summers
of 2018–2019. We define these impacts in a specific experiment by the differences between the mean seasonal concentrations
in this experiment and their corresponding values in the reference experiment. Before analyzing the mean seasonal impacts on
the concentrations of a specific OA, we first outline the spatial distributions of the mean seasonal concentrations of this OA
in the reference experiment. For this sensitivity study, we selected CSnI as the reference experiment since it was conducted
without any additional emissions of IVOCs and SVOCs. Following this, we evaluate the organic carbon (OC) concentrations
obtained from the individual experiments of this sensitivity analysis.

#### 3.2.1 Spatial distributions of POA and SOA

The spatial distributions of the mean seasonal POA concentrations in the reference experiment during the winters and summers are depicted in Figs. 4a and b, respectively. The mean winter POA concentrations usually range between  $0.1\text{--}9~\mu\mathrm{g}~\mathrm{m}^{-3}$ , with the highest values reaching in the Po Valley (Italy) and some areas of the Czech Republic and Poland. In contrast, the mean summer POA concentrations mostly reach up to  $0.3\text{--}0.5~\mu\mathrm{g}~\mathrm{m}^{-3}$ , except for the area of the Po Valley, where they reach up to  $1.5~\mu\mathrm{g}~\mathrm{m}^{-3}$ .

Regarding the mean seasonal impacts on POA concentrations, Figs. 4c and d show their spatial distributions during the winters and summers, respectively. Additionally, the values of the domain-averaged mean seasonal impacts on POA concentra-

Figure 4. Mean seasonal POA concentration (in  $\mu g m^{-3}$ ) in the reference experiment (CSnI) of the first sensitivity analysis during the winter (a) and summer (b) of 2018 and 2019. The difference between the mean seasonal POA concentration predicted in each individual experiment of the first sensitivity analysis and that in the CSnI experiment (in  $\mu g m^{-3}$ ) during the winter (c) and summer (d) of 2018 and 2019.

tions are provided in Table S9 in the Supplement. The distributions of the mean seasonal POA concentrations in CSwI, SSnI, and SSwI are practically identical to those in CSnI in both seasons, leading to the negligible mean seasonal impacts on POA concentrations in these experiments. However, this is an expected result since the same POA emissions were used in all these experiments, and POA is not affected by the choice of gas-phase mechanisms and the addition of IVOC emissions when using SOAP. On the other hand, the addition of SVOC emissions in CVb and CVa increases the mean seasonal POA concentrations

over the entire domain in both seasons. Compared to CSnI, the mean seasonal POA concentrations in CVb are higher on average by a factor of 1.65 and 1.74 in the winter and summer seasons, respectively, and in CVa by a factor of 1.66 and 1.80. These increases in the mean seasonal POA concentrations lead to the positive mean seasonal impacts, with spatial distributions that are largely consistent with the POA patterns observed in CSnI during both seasons. This resemblance in spatial patterns can be explained by the way SVOC emissions were scaled: POA emissions were used for most anthropogenic sources, while NMVOC emissions were used for diesel and gasoline vehicles, whose spatial distributions closely match those of POA emissions from the same vehicle categories. The most significant impacts in both experiments are observed in the Po Valley during both seasons, with the mean winter impacts reaching up to  $10 \,\mu \mathrm{g} \,\mathrm{m}^{-3}$  and the mean summer impacts reaching up to  $1 \,\mu \mathrm{g} \,\mathrm{m}^{-3}$ . The comparison of the domain-averaged seasonal impacts in these two experiments shows that the mean seasonal POA concentrations increase on average across the entire domain in CVa by  $0.02 \,\mu \mathrm{g} \,\mathrm{m}^{-3}$  in both seasons. This observed increase could be attributed to the fact that the additional aging of SOA gradually contributes to higher concentrations of the total OA, which subsequently shifts the thermodynamic equilibrium in the redistribution of POMsV between the gas and aerosol phase toward the aerosol phase.

Figures 5a and b illustrate the spatial distributions of the mean seasonal SOA concentrations in the reference experiment during the winters and summers, respectively. The mean winter SOA concentrations in most of the territory of the domain reach up to 0.2– $0.3 \, \mu \mathrm{g \, m^{-3}}$ . In contrast, the mean summer SOA concentrations range between 0.6– $1.6 \, \mu \mathrm{g \, m^{-3}}$  over most of the domain, with the highest values occurring in the southern areas of the Pannonian Basin and southern Germany.





As for the mean seasonal impacts on SOA concentrations, Figs. 5c and d show their spatial distributions during the winters and summers, respectively. Furthermore, Table S9 includes the values of the domain-averaged mean seasonal impacts on SOA concentrations ( $\triangle$  SOA). The distributions of the mean seasonal impacts in SSnI indicate that changing the mechanisms of gas-phase chemistry is almost not reflected in the mean winter SOA concentrations except for the central area of the Po Valley  $(\Delta SOA = 0 \mu g m^{-3})$ , while it causes a decrease in the mean summer SOA concentrations in most areas of the domain by 0.1–  $0.15 \ \mu g \ m^{-3}$  ( $\Delta SOA = -0.07 \ \mu g \ m^{-3}$ ). The addition of the IVOC emissions in CSwI and SSwI is manifested by the positive mean winter impacts over the entire domain, which are somewhat more pronounced in CSwI ( $\Delta$ SOA = 0.17  $\mu$ g m<sup>-3</sup>) than in SSwI ( $\Delta$  SOA = 0.12 µg m<sup>-3</sup>). On the other hand, even adding the IVOC emissions in SSwI during the summer seasons does not cause positive mean seasonal impacts over the whole domain ( $\Delta SOA = -0.01 \text{ ug m}^{-3}$ ), which is not the case in CSwI  $(\Delta SOA = 0.09 \text{ µg m}^{-3})$ . The simultaneous addition of the IVOC and SVOC emissions in CVb leads to a further increase in the mean seasonal impacts during both the winters ( $\triangle SOA = 0.49 \text{ µg m}^{-3}$ ) and the summers ( $\triangle SOA = 0.58 \text{ µg m}^{-3}$ ). The relatively smaller summer increase, compared to the winter increase, over regions such as the Po Valley, the Czech Republic, and the Pannonian Basin can be attributed to the seasonal reduction in IVOC and SVOC emissions from residential biomass burning. The additional aging of SOA in CVa raises the mean seasonal impacts even more during the winters ( $\Delta$  SOA = 0.62  $\mu g \, \mathrm{m}^{-3}$ ) and especially in the summers ( $\Delta \mathrm{SOA} = 1.17 \, \mu \mathrm{g} \, \mathrm{m}^{-3}$ ), resulting in the highest overall seasonal impacts among all the experiments examined in this sensitivity analysis. During the winters, the areas with the greater impacts in CVa include the Czech Republic and the Pannonian Basin, where the impacts reach 0.75-2 µg m<sup>-3</sup>. The highest impacts, reaching up to 4 ug m<sup>-3</sup>, occur in the Po Valley. During the summers, the areas with the higher impacts include the Czech Republic, southern

Figure 5. Same as Fig. 4 but for SOA.

Germany and Poland, northern Austria, and the Pannonian Basin, where they reach 1.25–2.5  $\mu g \, m^{-3}$ . Again, the Po Valley experiences the highest impacts, with values reaching up to 3.5  $\mu g \, m^{-3}$ .

Several recent studies have focused on modeling the influence of IVOC and SVOC emissions on OA concentrations over Central Europe using the CAMx model. Meroni et al. (2017) simulated OA in the Po Valley region on a domain with a horizontal resolution of  $5 \text{ km} \times 5 \text{ km}$  for February 2013. In one of their experiments, they used SOAP to represent OA chemistry, the CB05 mechanism (Yarwood et al., 2005) to simulate gas-phase chemistry, and included estimates of IVOC emissions, making their model setup closely aligned with that of our CSwI experiment. Taking into account the differences in

emission inventories, spatial resolution, and simulation period, the distribution of the mean monthly POA concentration in their experiment is qualitatively and quantitatively similar to the mean seasonal POA distribution in CSwI. In contrast, their mean monthly SOA concentrations, which reach a maximum of  $0.3 \, \mu \mathrm{g \, m^{-3}}$ , are lower than the mean seasonal SOA concentrations in CSwI. This discrepancy may stem, in part, from the fact that they estimated IVOC emissions from all sectors as  $1.5 \times \mathrm{POA}$ , i.e., using the non-source-specific parameterization proposed by Robinson et al. (2007). For instance, IVOC emissions from biomass burning, which dominate total IVOC emissions during winter, were presumably significantly underestimated in their experiment compared to our estimate of these emissions (4.5  $\times \mathrm{POA}$ ), which is based on recent biomass burning smog chamber experiments (Jiang et al., 2021; Ciarelli et al., 2017).

Ciarelli et al. (2017) used a modified version of the 1.5-D VBS scheme to model OA over the European domain with a horizontal resolution of  $0.25^{\circ} \times 0.25^{\circ}$  for the period between 25 February and 26 March 2009. To estimate the emissions of IVOCs and POM<sub>SV</sub>, they used parameterizations that align closely with those used in CVa. When considering the same aspects as in the previous comparison, it is evident that the modeled distributions of the mean concentrations of POA and SOA in their experiment show qualitative similarities to the mean winter concentrations of POA and SOA observed in CVa. The main quantitative differences between the distributions of the mean POA concentrations could be attributed to the use of different emission inventories. It is important to emphasize here that our assumption about the absence of SVOC emissions in the emission inventories used to prepare input emission data for our experiments (CAMS-REG-v4.2, REZZO, and ATEM; see Sect. 2.3) is only partially accurate. As noted by Kuenen et al. (2022), PM<sub>2.5</sub> emissions from small residential combustion, as reported in CAMS-REG-v4.2, include SVOC emissions for specific European countries. Italy is one such country, which suggests that the POM<sub>SV</sub> estimates provided in CVb and CVa likely led to an overestimation of POA over its territory. This overestimation, in turn, has naturally influenced (increased) SOA levels in this region, especially during the winter months. However, it is crucial to note that CAMS-REG-v4.2 does not account for SVOC emissions from Central European countries, just as REZZO and ATEM does not include them for the Czech Republic, thus justifying the use of POM<sub>SV</sub> estimates in CVb and CVa.

Jiang et al. (2021) modeled OA over the European domain with a horizontal resolution of  $0.25^{\circ} \times 0.125^{\circ}$  for the year 2011. Among the five experiments they performed, the SOAP and VBS\_3POA experiments are the most similar to CSwI and CVb, respectively, in terms of setting the OA chemistry module and the IVOC and POM<sub>SV</sub> parameterizations used. When examining the distributions of the mean winter and summer POA concentrations in the SOAP and VBS\_3POA experiments, a qualitative similarity can be observed with the corresponding distributions in CSwI and CVb. The quantitative differences between these distributions can be attributed to the use of different emission inventories since Jiang et al. (2021) employed the emission inventory TNO\_MACC-III (Kuenen et al., 2014), the predecessor of CAMS-REG-v4.2 that does not account for SVOC emissions. The more apparent differences in the mean winter POA concentrations between CVb and VBS\_3POA over Italy could be ascribed mainly to the above-mentioned overestimation of POA in CVb over this area. The distributions of the mean winter and summer SOA concentrations in the SOAP and VBS\_3POA experiments exhibit distinct patterns compared to those in CSwI and CVb, particularly over some regions in Germany, Poland, and the Czech Republic. These differences, which can reach up to about 2–3 μg m<sup>-3</sup> during both seasons, could be mainly caused by the use of different amounts of biogenic

emissions, especially monoterpene emissions. To substantiate this claim, it is noteworthy that Jiang et al. (2021) used the PSI model developed at the Laboratory of Atmospheric Chemistry of the Paul Scherrer Institute (Andreani-Aksoyoglu and Keller, 1995; Oderbolz et al., 2013; Jiang et al., 2019a) to estimate biogenic emissions. Furthermore, Jiang et al. (2019a) demonstrated that the biogenic emissions, particularly monoterpene emissions, estimated by the PSI model result in substantially higher SOA production than the biogenic emissions derived from the MEGAN model. Notably, the regions with the most pronounced differences in SOA production include, among others, the mentioned regions of Central Europe.

## 3.2.2 Comparison with measurements



Figure S4 in the Supplement compares the observed and modeled mean daily OC concentrations at the Prague–Suchdol and Košetice stations during the individual seasons. Figure S5 in the Supplement presents these comparisons at the stations used in both campaigns during their winter and summer phases. Figure 6 depicts the differences between the modeled and observed mean daily OC concentrations at the Prague–Suchdol and Košetice stations in the individual seasons. Figure 7 illustrates these differences at the stations used in both campaigns during their winter and summer phases. Table S10 in the Supplement offers the statistical comparison of the modeled and observed mean daily OC concentrations at the Prague–Suchdol and Košetice stations, while Table S11 in the Supplement provides the same statistical analysis for the stations used in both campaigns.

All these figures and the MB values in both tables indicate that the model in all the experiments generally underestimates the daily OC concentrations at all the stations, except for the Prague–Suchdol station, during all the periods of comparison. At the Prague–Suchdol station, the model behaves similarly, except in the CVb and CVa experiments during January–February (JF) 2018, when it slightly overestimates them (MB = 0.06 and 0.17  $\mu g \, m^{-3}$ , respectively). At the same time, several consistent patterns can be seen across all these comparisons.

**Figure 6.** Differences between the modeled and observed mean daily OC concentrations (in  $\mu$ g m<sup>-3</sup>) at the Košetice station during the winter (a), spring (b), summer (c), and autumn (d) of 2018 and 2019, and at the Prague–Suchdol station during January and February 2018 (e) and the spring of 2018 (f). The differences for all the model experiments of the first sensitivity analysis are shown. Blue dotted lines indicate the level of zero difference.

**Figure 7.** Differences between the modeled and observed mean daily OC concentrations (in  $\mu g \, m^{-3}$ ) at the Kosmos, Ropice, and Vrchy stations in the Třinecko area during the winter (**a-c**) and summer (**g-i**) phase of the campaign, and at the Švermov, Libušín, and Zbečno stations in the Kladensko area during the winter (**d-f**) and summer (**j-l**) phase of the campaign. The differences for all the model experiments of the first sensitivity analysis are shown. Blue dotted lines indicate the level of zero difference.

First, CSnI and SSnI typically underestimate these concentrations in a similar manner, with more pronounced differences during the summer comparison periods (i.e., the summer seasons at the Košetice station and the summer campaign phases in the Třinecko and Kladensko areas). Across all the stations and comparison periods, the mean percentage differences (MPDs) between the modeled and observed daily OC concentrations range from approximately -86.3 % to -36.8 % for CSnI and from -87.1 % to -36.8 % for SSnI. On average, the MPDs for SSnI are lower than those for CSnI by 1.5 % during the summer comparison periods and by 0.2 % during the other periods.



Second, CSwI and SSwI also tend to underestimate these concentrations in a similar manner, with more pronounced differences during the summer comparison periods. Across all the stations and comparison periods, the MPDs for CSwI and SSwI are slightly higher than those for their corresponding experiments without additional IVOC emissions, on average by 1.9 % for CSwI compared to CSnI and 1.3 % for SSwI compared to SSnI.

Third, CVb underestimates these concentrations even less than CSwI and SSwI, with MPDs that are, on average, 14.4 % higher than those for CSwI across all the stations and comparison periods, with the exception of a slight overestimation at the Prague–Suchdol station during JF 2018.

Finally, an additional average increase in MPD of 4.9 % relative to CVb across all the stations and comparison periods is reflected by the least underestimation overall in CVa, along with a more pronounced overestimation at the Prague–Suchdol station during JF 2018. This improvement is even more evident during the summer comparison periods, where the MPD increases by 6.6 % on average relative to CVb.

The trend in these patterns aligns with the seasonal distributions of POA and SOA described in Sect.3.2.1. Moreover, Tables S10 and S11 show a similar pattern of improving the model predictions of the daily OC concentrations in these experiments in terms of all the other statistical measures used.






Furthermore, the values of all the statistical metrics used demonstrate that the quality of the prediction of these concentrations is influenced by both the location within the domain and the period. The values of all the statistical metrics indicate that the model predictions of the mean daily OC concentrations at the Prague-Suchdol (NMSE = 45.4-67.1 %, IOA = 0.68-0.73, FAC2 = 66.7–80.0 %) and Košetice (NMSE = 21.0–57.8 %, IOA = 0.64–0.85, FAC2 = 65.5–81.9 %) stations (Table S10) are considerably more accurate in all the experiments during the winter seasons than those at the stations in the Kladensko and Trinecko areas during their winter phases (NMSE = 84.2–447. %, IOA = 0.46–0.58, FAC2 = 12.9–68.6 %; Table S11). These findings could be partly explained by the more pronounced differences between the modeled and measured wind speeds at the campaign stations. During the summer seasons, a similar pattern is observed in that the Košetice station (NMSE = 49.2–291.0 %, IOA = 0.36–0.52, FAC2 = 0–56.7 %) shows better prediction accuracy than the stations in the Kladensko and Třinecko areas (NMSE = 114.2–653.8 %, IOA = 0.36–0.49, FAC2 = 0–31 %; Table S11), which perform even more poorly than during the winter phases. Nevertheless, the model performance during the summer periods is consistently weak across all the stations and experiments, with a FAC2 exceeding 50 % only at the Košetice station in CVa. In order to make similar comparisons over the identical periods, we calculated all the metrics for the Košetice station corresponding to the periods of the individual phases of both campaigns, except for the summer phase in the Trinecko area due to missing data (Table S10). The comparisons of these metrics with those determined for the stations in both campaigns (Table S11) lead to the same conclusions that we stated above.

The comparison of NMSE, IOA, and FAC2 at the Prague–Suchdol station shows that the predictions of the mean daily OC concentrations in all the experiments are more accurate during the winter season than during the spring season (NMSE = 43.8–128.6%, IOA = 0.44–0.66, FAC2 = 47.8–69.6%; Table S10). Similar comparisons at the Košetice station also show that these concentrations are best predicted in all the experiments during the winter seasons. In contrast, as noted above, the predictions at this station are predominantly least accurate during the summer seasons.

The values of NMSE, IOA, and FAC2 at the stations used in both campaigns (Table S11) indicate a better prediction in all the experiments during the winter phases than in the summer phases. At the same time, they show that the winter phases are slightly better predicted in the Kladensko area, while the summer phases are slightly better predicted in the Třinecko area.

Finally, taking into account the values of all the mentioned statistical measures at all the stations considered here, as well as all the distributions of differences between the modeled and observed mean daily OC concentrations, we can state that daily OC concentrations are most accurately modeled in CVa, followed by CVb. In other words, the best modeled daily OC concentrations, although still underestimated, were achieved by simultaneously supplying estimates of both IVOC and SVOC

emissions to the simulation in which OA chemistry was handled by the 1.5-D VBS scheme with activated aging processes of POA and SOA from all anthropogenic sources as well as SOA from biogenic sources.

## 3.3 Sensitivity on chemical boundary conditions

This section presents the results of the second sensitivity analysis, in which the sensitivity experiments employed CBCs that differed from those prescribed in the reference experiments by modified gas-phase and additional aerosol species, as described in Sect. 2.4.2. To present and discuss these results, we follow a similar approach to that used in the previous sensitivity study. We first examine the spatial distributions of the mean seasonal impacts on the near-surface concentrations of POA and SOA in the experiments of this sensitivity analysis during both seasons, applying the same definition of these impacts as in the first sensitivity study. Finally, we evaluate the OC concentrations obtained from the individual experiments of this sensitivity analysis.

## 3.3.1 Spatial distributions of POA and SOA







Figures 8a and b show the spatial distributions of the mean seasonal concentrations of POA in both reference experiments (CSwI and CVb) during the winters and summers, respectively. As we showed in the first sensitivity analysis, these distributions in CSwI are almost identical to those in CSnI during both seasons (Figs. 4a and b). At the same time, the distributions in CVb exhibit similar spatial patterns to those in CSwI during both seasons, but they differ in magnitude. Specifically, the highest mean seasonal POA concentrations in CSwI reach up to 9  $\mu$ g m<sup>-3</sup> during the winters and up to 1.5  $\mu$ g m<sup>-3</sup> during the summers, whereas in CVb, these concentrations peak at 19  $\mu$ g m<sup>-3</sup> during the winters and 2.2  $\mu$ g m<sup>-3</sup> during the summers.

Figures 8c and d illustrate the spatial distributions of the mean seasonal impacts on POA concentrations during the winters and summers, respectively. Moreover, Table S9 includes the values of the domain-averaged mean seasonal impacts on POA concentrations ( $\triangle$  POA). When OA is represented as SOA at the boundaries of the model domain, the mean seasonal impacts on POA concentrations in Sp0s100 are minor in both seasons. In Vp0s100, these impacts during both seasons typically reach values up to 0.25 µg m<sup>-3</sup>, with somewhat more pronounced effects observed in the winters. This phenomenon may be attributed to the SOA supplied, which likely shifts the thermodynamic balance in the redistribution of POM<sub>SV</sub> between the gas and aerosol phases toward the aerosol phase. The subsequent increase in the share of POA in OA at the boundaries of the model domain in Sp50s50 and Sp100s0 (and analogously in Vp50s50 and Vp100s0) is manifested by a gradual rise in the mean seasonal impacts across the entire model domain during both seasons. The spatial distributions of the mean winter impacts in these model experiments feature asymmetric gradients predominantly oriented from the Alps toward the western, southern, and eastern boundaries. These gradients are further influenced locally by other mountain ranges, such as the High Tatras. The spatial distribution of the mean summer impacts exhibits a similar structure, with pronounced gradients extending toward all domain boundaries. As can be seen from the shape of these distributions and the values of  $\Delta POA$ , the increase in the mean seasonal impacts during both seasons, relative to those in Sp0s100 or Vp0s100 (depending on the OA module used), is consistently higher in the experiments where SOAP handles OA chemistry. This phenomenon can be attributed to the fact that, in Vp50s50 and Vp100s0, the total POA at the boundaries of the model domain is redistributed among the POA surrogate

Figure 8. Mean seasonal POA concentration (in  $\mu g \, m^{-3}$ ) in the reference experiments (CSwI and CVb) of the second sensitivity analysis during the winter (a) and summer (b) of 2018 and 2019. The difference between the mean seasonal POA concentration predicted in each individual experiment of the second sensitivity analysis and that in the corresponding reference experiment (CSwI or CVb) (in  $\mu g \, m^{-3}$ ) during the winter (c) and summer (d) of 2018 and 2019.

species (Table S6), of which only a portion (specifically PAP0 and PFP0) is treated as purely non-volatile. The remaining portion of this added POA can further partly evaporate inside the model domain, and part of this evaporated material can be further aged to form SOA. In contrast, in Sp50s50 and Sp100s0, the same total POA as in Vp50s50 and Vp100s0, respectively, enters the model domain from the boundaries, but it is treated entirely as non-volatile, preventing both its evaporation and

subsequent aging. The mean seasonal impacts and their domain-averaged values in both Sp50s50 and Vp50s50 are higher during the summer seasons ( $\Delta$  POA = 1.71 and 1.29  $\mu$ g m<sup>-3</sup>, respectively) than in the winter seasons ( $\Delta$  POA = 0.69 and 0.63  $\mu$ g m<sup>-3</sup>, respectively). Similarly, the mean seasonal impacts and their domain-averaged values in both Sp100s0 and Vp100s0 are also higher during the summer seasons ( $\Delta$  POA = 3.45 and 3.10  $\mu$ g m<sup>-3</sup>, respectively) than in the winter seasons ( $\Delta$  POA = 1.37 and 1.17  $\mu$ g m<sup>-3</sup>, respectively).

The spatial distributions of the mean seasonal concentrations of SOA in both reference experiments during the winters and summers are depicted in Figs. 9a and b, respectively. Similar to the spatial distributions of the mean winter POA concentrations, the mean winter SOA concentrations exhibit a similar spatial pattern in both experiments; however, they differ in magnitudes. While these concentrations usually range between  $0.1-1.25~\mu g~m^{-3}$  in CSwI, they vary between  $0.2-3.75~\mu g~m^{-3}$  in CVb. In contrast, the spatial distributions of the mean summer SOA concentrations in both experiments differ not only in size but also in the geographical locations of the areas with the highest values. In CSwI, the mean summer SOA concentrations typically range between  $0.4-1.75~\mu g~m^{-3}$ , with the highest values being reached mainly in the southern area of the Pannonian Basin. In CVb, these concentrations usually vary between  $0.8-3.5~\mu g~m^{-3}$ , with the highest values found in the Po Valley and southern Germany.

Finally, as for the mean seasonal impacts on SOA concentrations, Figs. 9c and d show their spatial distributions during the winters and summers, respectively. Furthermore, the values of the domain-averaged mean seasonal impacts on SOA concentrations are provided in Table S9 in the Supplement. When OA is represented as POA at the boundaries of the model domain, the mean winter impacts in Sp100s0 are negative, decreasing to -0.1  $\mu$ g m<sup>-3</sup>. In Vp100s0, they are also negative above most of the domain, dropping to -0.18  $\mu$ g m<sup>-3</sup>. On the contrary, the mean summer impacts in Sp100s0 are positive, reaching up to 0.5  $\mu$ g m<sup>-3</sup>. In Vp100s0, except for the Alps and High Tatras, they are similarly positive, reaching up to 0.5  $\mu$ g m<sup>-3</sup>.

Since the CBCs were the only factor varied between the simulations used to quantify these impacts (i.e., Sp100s0 vs. CSwI and Vp100s0 vs. CVb), the resulting differences in SOA concentrations can be directly attributed to modifications to the CBCs for both gas-phase and aerosol species. These modifications may affect both the oxidative environment, through species such as ozone, nitrogen oxides, carbon monoxide, and the hydroxyl radical, and the availability of direct SOA precursors such as toluene, xylene, and isoprene, helping to explain the spatial and seasonal variation observed. As mentioned earlier, the detailed composition of the two CBC sets is provided in Sect. 2.3 and Tables S1–S2.

The spatial distributions of the mean seasonal impacts on SOA concentrations in Sp50s50 and Sp0s100 are akin to the spatial distributions of the mean seasonal impacts on POA concentrations in Sp50s50 and Sp100s0, respectively (Figs. 8c and d), during both seasons. Similarly, the spatial distributions of the mean seasonal impacts on SOA concentrations in Vp50s50 and Vp10s100 resemble those for POA concentrations in Vp50s50 and Vp100s0, respectively, during both seasons. This resemblance in the spatial patterns may be partially explained by taking into account the volatility characteristics of the OA surrogate species at the boundaries of the model domain. As indicated in Tables S4–S6, the POA and SOA fractions of OA are, in both seasons, predominantly redistributed into non-volatile surrogate species and those with low volatility. As noted in the methodology, POA is fully non-volatile in the experiments where SOAP is applied. These components either reside entirely (in the case of non-volatile species) or predominantly in the aerosol phase, in which they are transported across the domain. Given that all the

Figure 9. Same as Fig. 8 but for SOA.

simulations are driven by the same meteorological conditions, a similar transport behavior of the non-volatile and low-volatility fractions of POA and SOA is expected, further contributing to the observed resemblance in the spatial patterns.

The domain-averaged values of the mean seasonal impacts on SOA concentrations in Sp50s50 and Vp50s50 are higher in the summers ( $\Delta$  SOA = 1.22 and 1.15  $\mu$ g m<sup>-3</sup>, respectively) than in the winters ( $\Delta$  SOA = 0.55 and 0.41  $\mu$ g m<sup>-3</sup>, respectively). Similarly, the domain-averaged values of these mean seasonal impacts in Sp0s100 and Vp0s100 are also higher in the summers ( $\Delta$  SOA = 2.27 and 2.03  $\mu$ g m<sup>-3</sup>, respectively) than in the winters ( $\Delta$  SOA = 1.15 and 0.83  $\mu$ g m<sup>-3</sup>, respectively). However, the mean seasonal impacts on SOA concentrations in these simulations are not consistently higher in the summers across the

entire domain. In some areas localized within the southeastern part of the domain, the impacts are actually higher during the winters, by up to  $0.1 \ \mu g \ m^{-3}$  in Sp50s50 and Vp50s50, and by up to  $0.5 \ \mu g \ m^{-3}$  in Sp0s100 and Vp0s100. The observed seasonal differences in the mean seasonal impacts are driven by a combination of the seasonal variation in the mean monthly OA concentrations at the boundaries, which affects all four simulations, and the redistribution of these concentrations among (1) the SOA surrogate species in Sp0s100 and Vp0s100, and (2) the POA and SOA surrogate species in Sp50s50 and Vp50s50.

#### 3.3.2 Comparison with measurements





The observed and modeled mean daily OC concentrations at the Prague–Suchdol and Košetice stations in the individual seasons are compared in Fig. S6 in the Supplement. Figure S7 in the Supplement shows these comparisons at the stations employed in both campaigns during their winter and summer phases. Figure 10 depicts the differences between the modeled and observed mean daily OC concentrations at the Prague–Suchdol and Košetice stations in the individual seasons. Figure 11 illustrates these differences at the stations used in both campaigns during their winter and summer phases. The statistical comparison of the modeled and observed mean daily OC concentrations at the Prague–Suchdol and Košetice stations is provided in Table S12 of the Supplement. Tables S13 and S14 in the Supplement provide the same statistical analysis for the stations used in both campaigns.

All these figures and the MB values in the tables show that, apart from the slight overestimation of the mean daily OC concentrations observed at the Košetice station in CVb during JF 2018, these concentrations are underestimated in both reference experiments, as discussed in Sect. 3.2.2. At the same time, they demonstrate that incorporating OA into the CBCs reduces these underestimations in all cases where they occur in the reference experiments, resulting in improved model predictions of the analyzed concentrations. In contrast, at the Košetice station during JF 2018, the incorporation of OA into the CBCs in the experiments using the 1.5-D VBS scheme exacerbates the existing overestimation of the mean daily OC concentrations observed in CVb, leading to degraded model predictions of the analyzed concentrations in these experiments. Both the reductions in the underestimations and the exacerbations of the overestimations become more pronounced with the increasing shares of POA

Figure 10. Same as Fig. 6 but for the experiments of the second sensitivity analysis.

Figure 11. Same as Fig. 7 but for the experiments of the second sensitivity analysis.



in OA at the boundaries of the model domain. Moreover, in some cases, the incorporation of OA into the CBCs even leads to slight overestimations of the mean daily OC concentrations, which were underestimated in the reference experiments, e.g., at the Košetice station in (1) Vp0s100, Vp50s50, and Vp100s0 during the winter seasons (Figs. 10a and S6a, Table S12), and (2) Sp100s0 and Vp100s0 during the summer seasons (Figs. 10c and S6c, Table S12).

Tables S12–S14 further indicate that increasing the share of POA in OA at the boundaries of the model domain tends to improve the majority of the remaining statistical metrics (i.e., RMSE, NMSE, IOA, and FAC2) across the stations and observational periods. This trend aligns with expectations, at least during the winter and summer seasons, given the predominantly underestimated concentrations in the reference experiments and the spatial distributions of OA (Fig. S8 in the Supplement), the components of which we detailed in the previous subsection. The FAC2 values reveal that the most pronounced improvements in the modeled mean daily OC concentrations occur mainly during the summer seasons, particularly at the stations in the Kladensko area and at the Košetice station, where they increased from 0–3.3 % to 50–100 % and from 0–16.4 % to 57.7–93.3 %, respectively.

The fact that the improvement or deterioration in the modeled mean daily OC concentrations resulting from the addition of OA at the boundaries of the model domain differs between the stations and seasons analyzed can be attributed to the combined influence of several interacting factors that vary both spatially and temporally. These include (1) the annual variation in the mean monthly concentrations of the total OA prescribed at the boundaries of the model domain, (2) the seasonal variation in how these concentrations are redistributed into the POA and SOA surrogate species, (3) changes in atmospheric conditions

that affect the transport and chemistry of OA (e.g., wind patterns and temperature), and (4) spatial and temporal variability in anthropogenic and biogenic emissions inside the model domain.

Finally, it is worth noting that the redistribution scenarios treating total OA at the boundaries of the model domain as entirely POA or entirely SOA were not intended to represent realistic conditions, but were designed as bounding cases to assess the sensitivity of the model to the unknown OA composition in the EAC4 dataset by exploring the maximum plausible range of impacts on the modeled mean daily OC concentrations. Although Sp100s0 and Vp100s0 produced the mean daily OC concentrations that most closely matched the observations in this sensitivity study, they assumed OA to be entirely composed of POA at the boundaries of the model domain, which is highly unrealistic. As already noted in the Introduction, Chen et al. (2022) found that SOA dominates the organic aerosol fraction of PM<sub>1</sub> across Europe (ranging from 47.3 % to 100 %), indicating that a significant SOA component may be expected in real boundary conditions. While their results pertain to PM<sub>1</sub>, they suggest that the improvements in the modeled mean daily OC concentrations obtained in Sp50s50 and Vp50s50, or in simulations falling between Sp50s50 and Sp0s100 and between Vp50s50 and Vp0s100, may more realistically reflect the influence of OA composition at the boundaries of the model domain.

## 4 Conclusions







This study presents a comprehensive examination of OA modeling over Central Europe, using two sensitivity analyses to explore the influence of IVOC and SVOC emissions and the OA composition in chemical boundary conditions. The first sensitivity analysis showed that the inclusion of source-specific and non-source-specific estimates of IVOC and SVOC emissions substantially affects the modeled POA and SOA concentrations. A comparison of modeled daily OC concentrations with measurements at stations in the Czech Republic showed that the model reproduced these concentrations most accurately at both rural and urban stations when the OA chemistry was controlled by the 1.5-D VBS scheme with activated aging processes for POA and SOA from all anthropogenic sources and for SOA from biogenic sources, in combination with the inclusion of the source-specific and non-source-specific IVOC and SVOC emissions (the CVa experiment). In this experiment, the domain-averaged mean seasonal concentrations of POA increased by approximately  $0.78 \,\mu \mathrm{g \, m^{-3}}$  in the winter seasons and  $0.17 \,\mu \mathrm{g \, m^{-3}}$  in the summer seasons, while those of SOA rose by about  $0.62 \,\mu \mathrm{g \, m^{-3}}$  and  $1.17 \,\mu \mathrm{g \, m^{-3}}$ , respectively. These increases contributed to improved model performance, especially during the winter periods, and underscore the importance of including IVOC/SVOC emissions and aging processes in OA modeling.

Under the CVa configuration, the model performance was generally better during the winter periods than in the summer periods, and the rural stations were better predicted than the urban stations in both seasons. During the winter periods, NMSE ranged from 45.4–220.3 %, IOA from 0.50–0.73, and FAC2 from 37.1–80.0 % at the urban stations, while at the rural stations, NMSE ranged from 21.0–160.7 %, IOA from 0.53–0.85, and FAC2 from 22.6–81.9 %. In the summer periods, the model accuracy consistently declined, with NMSE ranging from 117.5–186.6 %, IOA from 0.43–0.48, and FAC2 from 0.0–6.9 % at the urban sites, and NMSE from 49.2–154.7 %, IOA from 0.46–0.52, and FAC2 from 6.7–56.7 % at the rural sites. The better performance in the winter periods likely reflects the higher amounts of IVOC and SVOC emissions during this season, which

may be more reliably captured by the model. Nevertheless, it should be noted that these findings are based on a relatively small number of urban and rural stations, and differences in the temporal coverage of the measurements across the stations may influence the results.

The second sensitivity analysis, based on simulations using the CSwI and CVb experiments as the reference experiments, examined the impact of the organic aerosol composition in the chemical boundary conditions. Including OA at the model domain boundaries generally improved the modeled daily OC concentrations, with the most pronounced improvements occurring during the summer periods. These improvements were most substantial when the boundary OA was assumed to consist entirely of POA (Sp100s0 and Vp100s0), but these configurations were intended as bounding cases rather than realistic scenarios. More plausible simulations, such as Sp50s50 and Vp50s50 (with 50 % POA and 50 % SOA in the boundary OA), or cases between them and those using purely SOA-based boundary conditions, also showed improvement in the modeled daily OC concentrations, although generally less pronounced, and likely offer a more realistic reflection of the influence of OA at the domain boundaries. The FAC2 values in these experiments revealed that the most pronounced improvements occurred mainly during the summer periods, particularly at the stations in the Kladensko area, where values increased from 0–3.3 % to 50–90 %, with slightly better performance at the rural station in Zbečno, and at the Košetice station, which is also a rural site, where values increased from 0–16.4 % to 57.7–92.3 %. These findings also underscore the importance of including OA in the boundary conditions for accurate OA modeling, particularly during the summer seasons.





Because the CVa experiment provided the best overall model performance in the first sensitivity analysis, it can be considered the most suitable setup for modeling OA in Central Europe. In contrast, the second analysis demonstrated that the inclusion and composition of OA in the boundary conditions is most influential during summer, especially at rural sites. Although the Vp0s100, Vp50s50, and Vp100s0 experiments in the second sensitivity analysis used the same setup as the CVb experiment, which differs from CVa in the treatment of SOA aging from biogenic and biomass burning sources, qualitatively similar results would likely hold if CVa were used as the reference experiment. Taken together, these results highlight that both the inclusion of IVOC and SVOC emissions and the application of the 1.5-D VBS scheme with aging from all OA sources, as well as boundary OA, can substantially influence model performance, with their relative importance varying by season and location. Therefore, combining the setup used in CVa with realistic OA boundary conditions would likely offer the most robust modeling strategy.

While this study provides several important insights, some limitations remain. In particular, the number and duration of available measurement campaigns constrain the spatial and temporal representativeness of model evaluation. Moreover, several model uncertainties warrant further investigation, including the source-specific parameterizations for IVOC and SVOC emissions, the volatility distributions of SVOCs, the rate constants used in aging processes, and the emissions of BVOCs. Future work should also focus on increasing the temporal resolution of chemical boundary conditions and using input data that directly distinguish between POA and SOA. Additionally, increasing the horizontal resolution of the model domain may help mitigate wind speed overestimations and improve spatial accuracy in urban areas and regions with complex terrain.

Code and data availability. CAMx version 7.10 is available at http://camx-wp.azurewebsites.net/download/source (Ramboll, 2021). WRF version 4.2 can be downloaded from https://github.com/wrf-model/WRF/releases (WRF, 2020). MEGAN version 2.10 can be obtained from https://bai.ess.uci.edu/megan/data-and-code/megan21 (Guenther et al., 2014). The FUME emission model can be found at https://doi.org/10.5281/zenodo.10142912 (Belda et al., 2023). OC measurements at the Košetice station can be obtained from the EBAS database available at https://ebas-data.nilu.no/default.aspx (EBAS, 2025). All meteorological data used in the paper can be obtained from the Czech Hydrometeorological Institute (CHMI; https://www.chmi.cz). The CAMS global reanalysis (EAC4) monthly averaged fields can be downloaded from https://ads.atmosphere.copernicus.eu/datasets/cams-global-reanalysis-eac4-monthly?tab=download (ADS, 2025). The Czech REZZO and ATEM emission data can be obtained on request from their publishers, the CHMI and the Studio of Ecological Models (https://www.atem.cz), respectively. The complete model configuration and all the simulated data (1-dimensional hourly data) used for the analysis are stored at the Department of Atmospheric Physics of the Charles University data storage facilities (about 3TB) and are available upon request from the main author.

Author contributions. LB performed the simulations, analyzed and validated the modeled data, and wrote most of the text. PH and JK contributed to setting up the model simulations and writing the text. JP and OV helped with the methodology, and PV carried out and processed the OC measurements at the Prague–Suchdol station.

Competing interests. The authors declare that they have no conflict of interest.



Acknowledgements. This work has been funded by the Czech Technological Agency (TACR) grant No.SS02030031 ARAMIS (Air Quality Research Assessment and Monitoring Integrated System) and partly by the project of the Charles University SVV no. 260709 and by the Project OP JAK "Natural and anthropogenic georisks" CZ.02.01.014/0022\_008/0004605. We also acknowledge the CAMS-REG-v4.2 emission data and the CAMS global reanalysis (EAC4) monthly averaged fields provided by the Copernicus Atmosphere Monitoring Service; the REZZO dataset, the OC measurements from the campaigns in the Třinecko and Kladensko areas, and all the meteorological data provided by the Czech Hydrometeorological Institute; the OC measurements at the Prague–Suchdol station provided by the Institute of Chemical Process Fundamentals of the Czech Academy of Science; the ATEM Traffic Emissions dataset provided by ATEM (Studio of Ecological Models), and the ERA-Interim reanalysis provided by the European Centre for Medium-Range Weather Forecast. We also acknowledge the providers of the EBAS database, the Norwegian Institute for Air Research (NILU), and the Co-operative Programme for Monitoring and Evaluation of the Long-range Transmission of Air Pollutants in Europe (EMEP) for providing OC measurements at the Košetice station.

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
