# Peer review of "Modeling organic aerosol over Central Europe: uncertainties linked to different chemical mechanisms, parameterizations, and boundary conditions"

_EGUsphere, 2025_

## Author Comment (AC1)

**Authors' response to Anonymous Referee #1**

**on review of** *"Modeling organic aerosol over Central Europe: uncertainties linked to different chemical mechanisms, parameterizations, and boundary conditions"*

**by Lukáš Bartík et al. (ecusphere-2025-167 )**

Dear Anonymous Referee #1,

We sincerely thank you for the time and effort you dedicated to reviewing our manuscript, and for your constructive and insightful comments. Please find below our detailed, point-by-point responses (in black) to the comments you provided (in blue).

The manuscript "Modeling organic aerosol over Central Europe: uncertainties linked to different chemical mechanisms, parameterizations, and boundary conditions" by Bartík et al. combines CAMx model simulations with observations from sites located in Czech Republic. They investigate the sensitivity of organic aerosol concentrations simulated with CAMx on assumptions regarding IVOC and SVOC emissions and model boundary conditions. The topic fits within the scope of ACP. The model evaluation and sensitivity analysis presented in the manuscript are valuable work towards improving model representation of organic aerosols. However, the manuscript contains shortcomings in the description of methods and results and these should be addressed before the manuscript can be recommended for publication.

**Major comments**

1. The main aspects of the model related to the presented analysis should be included for the reader to be able to understand the work. I find that following aspects of the model should be described better:

- L121-126: Why were two chemistry mechanisms used? Could the authors please explain here some basics of what kind of chemistry these two mechanisms include and what are the main similarities/differences between them or otherwise explain the use of two mechanisms. For example, do the two mechanisms include essentially different precursors and/or reaction products?

  We agree that the original version of the paragraph lacked sufficient detail to clarify the rationale and differences between the two gas-phase chemistry mechanisms. In the revised manuscript, we expanded the paragraph to explain the key chemical processes covered by both mechanisms. These include photolytic reactions and oxidation by hydroxyl radicals, nitrate radicals, and ozone, as well as the formation and reactions of hydroperoxyl and organic peroxy radicals. We also mentioned that the CB6r5 mechanism is a lumped-structure mechanism that groups volatile organic compounds (VOCs) based on their chemical structure and bond type, while explicitly treating selected compounds, such as isoprene, formaldehyde, and acetaldehyde. In contrast, SAPRC07TC applies lumping primarily based on VOC reactivity and includes a larger number of VOCs and their oxidation products in explicit form.

  We also added a new paragraph to explain the reasons for using both mechanisms. These were selected to evaluate how differences in gas-phase chemistry formulations affect secondary organic aerosol (SOA) production, while keeping the SOA treatment unchanged. Additionally, we clarify that their use reflects practical constraints in the available configurations of the CAMx model, as only certain combinations of gas-phase mechanisms and OA modules are supported without requiring modifications to the model code.

- L127-134: Since aerosols are in focus in this study, please describe the basics of aerosol representation in the model, e.g.: What aerosol dynamics processes are included? Is condensation calculated based on equilibrium partitioning or some other way? It is said that ISORROPIA is used to predict composition and physical phase of inorganic aerosols. Are the organic and inorganic aerosols assumed externally mixed?

We agree that the basics of aerosol representation in the model should be provided and have accordingly revised the relevant paragraph in the manuscript. We now specify that we selected the coarse/fine (CF) aerosol scheme (Ramboll, 2020) to couple aerosol processes with gas-phase chemistry. This choice reflects the fact that it is the only scheme in CAMx that supports all combinations of gas-phase mechanisms and organic aerosol modules utilized in our experiments, which are described in Sect. 2.4 of the revised manuscript.

We clarify that the CF scheme divides the aerosol size distribution into two static, non-interacting modes (fine and coarse), within which aerosols are treated as internally mixed and monodisperse in size. Primary aerosol species can be represented in one or both modes, whereas all secondary aerosol species are modeled exclusively in the fine mode. We also distinguish the treatment of coarse- and fine-mode aerosols. Coarse-mode aerosol species are treated as non-volatile, chemically inert, and subject only to emission, transport, and removal by dry and wet deposition. In contrast, while all fine-mode aerosol species undergo the same physical processes, many of them can also participate in gas–particle partitioning, which is calculated based on the thermodynamic equilibrium assumption and applied separately for inorganic and organic aerosol species.

We also clarify that ISORROPIA version 1.7 is used to predict the composition and physical phase of inorganic aerosols, and that it models the sodium–ammonium–chloride–sulfate–nitrate–water system, including the mutual deliquescence behavior of multicomponent salt particles. We retained and clarified that one of two modules—SOAP (Secondary Organic Aerosol Processor) version 2.2 or 1.5-D VBS (1.5-dimensional Volatility Basis Set)—is used in CAMx version 7.10 to control organic gas–particle partitioning and oxidation chemistry. Lastly, we also retained and clarified that the CF scheme includes aqueous aerosol formation in resolved cloud water, calculated using a modified version of the RADM (Regional Acid Deposition Model) aqueous chemistry algorithm, which accounts for aqueous SOA formation from water-soluble precursors such as glyoxal, methylglyoxal, and glycolaldehyde.

We hope that these clarifications enhance understanding of aerosol treatment in our model simulations and address the referee's specific questions, which we summarize below along with our responses:

- What aerosol dynamics processes are included?

  Aerosol dynamics depend on particle size. Coarse-mode aerosol species are treated as non-volatile, chemically inert, and subject only to emission, transport, and removal by dry and wet deposition. All fine-mode aerosol species undergo the same physical processes as coarse-mode species, and many of them can also participate in gas–particle partitioning, depending on their composition.

- Is condensation calculated based on equilibrium partitioning or some other way?

  Yes, condensation—as part of gas–particle partitioning—is calculated based on the thermodynamic equilibrium assumption, using ISORROPIA for inorganics and either SOAP or 1.5-D VBS for organics.

- Are organic and inorganic aerosols assumed externally mixed?

  No, organic and inorganic aerosol species are treated as internally mixed within each size mode in the CF scheme. However, gas–particle partitioning of fine-mode aerosol species is applied separately for inorganic and organic species.

- L159-160: What are these concentrations of the chemical species based on?

The default chemical boundary conditions consist of time-space invariant concentrations of ozone and its precursors, including several reactive nitrogen compounds and non-methane volatile organic compounds (NMVOCs). Their values reflect typical background concentrations over Europe and were derived from simulations performed by Huszar et

al. (2020) over a large European domain with a horizontal resolution of 27 km. This information has been added to the penultimate paragraph of Sect. 2.3 in the revised manuscript.

- Table S1: The SOAP and VBS schemes seem to consider monoterpenes and sesquiterpenes but these species do not seem to be included in the boundary conditions. If that is the case, why is it so? At least the overall picture of what is and what isn't included in the boundary conditions should be described in the main text.

Indeed, monoterpenes and sesquiterpenes are not included in the default chemical boundary conditions (CBCs), as these are also not available in the EAC4 dataset. For consistency, we omitted them from the default CBCs as well. On the other hand, the most important biogenic volatile organic compound, isoprene, is included in both datasets.

Regarding the overall picture, we agree that a brief summary and comparison of the species included in each CBC set should be provided in the main text. To address this, we explicitly stated in the penultimate paragraph of Sect. 2.3 in the revised manuscript that the default CBCs contain ozone and its precursors, including several reactive nitrogen compounds and NMVOCs, with the full list provided in Table S1 of the revised Supplement. In the final paragraph of the same section, we added that the EAC4 CBCs include some gas-phase species that are absent from the default CBCs and vice versa, with concrete examples. More importantly, we also explicitly stated that the EAC4 CBCs differ fundamentally from the default CBCs by incorporating aerosol species, namely sea salt, dust, sulfate, hydrophobic black carbon, hydrophilic black carbon, hydrophobic organic matter, and hydrophilic organic matter. The full list of gas-phase and aerosol species included in the EAC4 CBCs is provided in Table S2 of the revised Supplement. We hope that these additions offer a clearer and more complete understanding of the CBCs used in our simulations.

- L180-183: Authors should clarify here that the aging is included in CVb for POA and anthropogenic SOA. In appendix on L574 it is stated that "The gas-phase hydroxyl radical reaction rates for the chemical aging of POA and anthropogenic SOA, except those originating from biomass burning, are assumed to be 4×10−11 and 2×10−11 cm3 molecule−1 s−1, respectively. In contrast, the chemical aging of biogenic SOA and SOA originating from biomass burning (both anthropogenic and biogenic) is disabled." There authors should make it clear that this disabling of aging refers to the reference run, not for all runs with VBS scheme.

We agree that the distinction regarding which aging processes are included in the CVb and CVa experiments was not sufficiently clear. In response, we revised the main text to explicitly state that the CVb experiment uses the default configuration of the 1.5-D VBS scheme, which includes the chemical aging of POA and anthropogenic SOA (excluding biomass burning), while aging of biogenic and biomass-burning-derived SOA remains disabled. We also clarified that CVa builds upon this configuration by enabling the additional aging of biogenic and biomass-burning SOA using the same OH reaction rate as for anthropogenic SOA.

Additionally, in response to your third minor comment, we have relocated the content previously in Appendix A to a new subsection (Sect. 2.2) in the revised manuscript. In this updated section, we clarified that the disabling of aging for biogenic and biomass-burning SOA is part of the default configuration of the 1.5-D VBS scheme and that this setup was specifically applied in the CVb experiment. By contrast, the CVa experiment extends this default configuration by activating the additional aging pathways for these SOA types. We hope these clarifications will help the reader better understand the chemical aging configurations used in both experiments and their implications for OA formation.

- L185-190: Why were different emissions used for simulations with different SOA mechanisms? Also, were all POA emissions really replaced with POM_SV emissions? Does that mean that all POA was assumed to be semivolatile? Use of POM_SV in the model is one of the main focus points in the manuscript and Table 2 lists parameterizations used for these, but explanation of how these POM_SV are treated in the model is missing. Does all POM_SV have same volatility in the model?

Before directly addressing the first question regarding the use of different emissions across simulations, we believe it is important to first clarify the definition of $POM_{SV}$ and several relevant aspects of the 1.5-D VBS scheme.

To address this, we have modified the definition of $POM_{SV}$ in the introduction of the revised manuscript to clarify that it refers to primary organic matter that spans both the semi-volatile and lower-volatility parts of the volatility spectrum. We acknowledge that our previous formulation may have been imprecise in conveying what we intended to express.

We have also made changes to the description of the 1.5-D VBS scheme to better clarify several of its aspects. As mentioned in our previous response, this updated content (originally part of Appendix A) is now presented in Sect. 2.2 of the revised manuscript. In particular, we have not only explained that the basis sets comprise five volatility bins with saturation concentrations of $C^0 = \{10^{-1}, 10^0, 10^1, 10^2, 10^3\}$ $\mu g\,m^{-3}$ at 298 K, but also highlighted that, although the properties of the surrogate species in the lowest volatility bin were estimated assuming $C^0 = 10^{-1}\,\mu g\,m^{-3}$, they in fact represent all OA of a given type with $C^0 \leq 10^{-1}\,\mu g\,m^{-3}$ and are treated as non-volatile. This means that whenever the gas-phase surrogate species in the lowest volatility bin is produced in any of the basis sets via chemical aging, it is assumed to immediately condense into its corresponding particle-phase surrogate species, which is treated as non-volatile and does not evaporate.

In this newly integrated section, we also provide a more precise explanation of how the 1.5-D VBS scheme treats POA and IVOC emissions, particularly in contrast to SOAP. The updated text more clearly describes that, unlike SOAP, which treats anthropogenic IVOCs using a single surrogate species, the 1.5-D VBS scheme uses four source-specific surrogate species for IVOC emissions, corresponding to gasoline vehicles, diesel vehicles, other anthropogenic sources, and biomass burning. Similarly, we clarified that POA emissions are not mapped to a single non-volatile species (as in SOAP), but are instead allocated to one of three basis sets representing freshly emitted OA, depending on the emission source. Within the assigned basis set, these POA emissions are further redistributed across all volatility bins using source-specific volatility distribution factors. The scheme distinguishes between POA emissions from gasoline vehicles, diesel vehicles, meat cooking, other anthropogenic sources, and biomass burning, applying a separate set of volatility distribution factors to each of these source categories.

We hope that these revisions to the manuscript provide a much clearer understanding of what we originally intended to express under the term $POM_{SV}$, as well as of the representation of the 1.5-D VBS scheme and its inputs. With this clarification in place, we now return to the original question regarding the use of different emissions across simulations.

To address this question, we have clarified this point in the revised manuscript. Specifically, we have expanded the explanation in Sect. 2.3 of the revised manuscript (formerly Sect. 2.2) to emphasize that the traditional POA emissions from inventories were retained only in experiments using the SOAP mechanism, where POA is treated as non-volatile. In these experiments, we assumed—consistent with the approach adopted in many previous studies cited in the Introduction—that traditional POA emissions do not account for missing SVOCs. However, as described in Sect. 2.2, POA emissions in the 1.5-D VBS scheme are redistributed across all the volatility bins within the appropriate basis set, based on their source, and should therefore include the missing SVOCs. Consequently, in the experiments employing the 1.5-D VBS scheme, we replaced the traditional POA emissions with those for $POM_{SV}$ to ensure inclusion of the missing SVOC fraction.

In response to the second question — *"Also, were all POA emissions really replaced with $POM_{SV}$ emissions?"* — we confirm that the answer is yes. In all experiments utilizing the 1.5-D VBS scheme, traditional POA emissions were fully replaced with the corresponding $POM_{SV}$ emissions, using the parameterizations provided in Table 2. To clarify this implementation, we have expanded the final paragraph of Sect. 2.3 in the revised manuscript, where we now describe the specific parameterizations used for each source category, including gasoline vehicles, diesel vehicles, residential biomass burning, and other anthropogenic sources.

In response to the third question — *"Does that mean that all POA was assumed to be semivolatile?"* — we clarify that this is not the case. As explained earlier in this response and detailed in the newly inserted Sect. 2.2 of the revised manuscript, a portion of POA—and therefore also a portion of $POM_{SV}$ in our case—is still treated as non-volatile. Specifically, emissions assigned to the lowest volatility bin are assumed to irreversibly partition to the particle phase, representing non-volatile material within the 1.5-D VBS scheme.

In response to the remainder of the comments regarding the treatment of $POM_{SV}$ in the model and its volatility distribution, we have clarified these points in the revised manuscript. Specifically, at the end of the penultimate paragraph of Sect. 2.3, we have added a statement noting that, apart from accounting for the missing SVOCs, $POM_{SV}$ emissions are otherwise treated identically to POA emissions within the 1.5-D VBS scheme. This clarifies that $POM_{SV}$ is not treated as having a single volatility, but is handled within the existing volatility-resolved framework of the scheme. Furthermore, in the final paragraph of Sect. 2.3, we now provide details on the volatility distribution factors used to allocate $POM_{SV}$ emissions from each source category to the volatility basis sets. These allocation factors are also listed in the newly added Table S3 in the revised Supplement.

- L560-563: What does "more-volatile" and "less-volatile" mean concretely in terms of volatilities?

  The model represents the volatility of these substances using saturation concentrations ($C^0$). At a temperature of 300 K, the more-volatile and less-volatile products have the following $C^0$ values: (1) for anthropogenic precursors, $C^0$ = 14 and 0.31 μg/m³, respectively; and (2) for biogenic precursors, $C^0$ = 26 and 0.45 μg/m³, respectively (Ramboll, 2020). These saturation concentration values have now been explicitly included in the newly integrated Sect. 2.2 of the revised manuscript, which incorporates the content previously presented in Appendix A.

2. Description of the observational data used for the model evaluation would need more information:

- A map showing the locations of the observational sites would be helpful for a reader. This could be a separate map or the locations could be marked in e.g. in the Fig. 1.

  We agree that adding a map showing the locations of the measuring stations would be helpful for readers. In response, we have added an additional panel (b) to Fig. 1 in Sect. 2.1 of the revised manuscript, showing the area of the Czech Republic together with the locations of all the stations used for validation.

- L237-238: According to the Table S6, the length of each of these measurement campaigns was only about one month. Please mention that in the main text.

  We acknowledge that the approximate duration of the measurement campaigns should be stated in the main text. Accordingly, we revised the sentence in Sect. 2.5 of the revised manuscript (formerly Sect. 2.4 in the original manuscript) to note that each campaign phase lasted approximately one month. We also updated the reference to Table S7 in the revised Supplement (formerly Table S6 in the original Supplement), which provides the exact schedules.

- L250-252: Please mention how long time period was considered from these data.

  We recognize that the time period covered by the observational data should be clearly stated. Accordingly, we have specified in Sect. 2.5 of the revised manuscript that these OC measurements used for validation were taken at the Košetice station from 1 January 2018, 02:00 UTC, to 31 December 2019, 02:00 UTC. We also noted that the daily OC concentrations were derived as 24-hour averages that follow the station's sampling schedule (i.e., from 02:00 UTC on a given day to 02:00 UTC on the following day). In addition, we revised the paragraph describing the observational data at the Prague–Suchdol station to clarify that sampling initially started at 09:00 UTC, but from 27 March 2018 onward, the start time was adjusted to 08:00 UTC. This information has now been explicitly included in the revised manuscript to improve transparency and accuracy.

In addition to the changes discussed in the three points above, we also revised the final two paragraphs of Sect. 2.5, *"Validation"* (formerly Sect. 2.4), to improve clarity and precision. In the penultimate paragraph of the original version, we incorrectly implied that meteorological variables were not measured at the Prague–Suchdol station. In the revised manuscript, we clarify that Prague–Suchdol is an air quality station where accompanying meteorological measurements are indeed conducted. However, the relevant meteorological data from this site were not included in the dataset provided to us, which was limited to professional meteorological stations. For this reason, we used data from the nearby Prague–Kbely station—a professional station—which we selected as a representative site for Prague–Suchdol.

In the final paragraph, we clarified the rationale for comparing daily OC concentrations—namely, to ensure at least consistency in the duration of sampling periods across all the stations, as the longest period among them was 24 hours (at the Prague–Suchdol station), even though the sampling windows differ between sites. We also specified that CAMx was configured to output hourly averaged concentrations, which allowed us to construct daily model outputs that matched the sampling periods at each station. Additionally, we made clear that for the meteorological evaluation, only the mean daily modeled and observed values from the days with available OC measurements were used, with each daily value constructed to follow the station-specific 24-hour sampling windows.

Importantly, we would like to point out that in the case of the analysis for the Prague–Suchdol and Košetice stations, we made a mistake by comparing the measured daily OC concentrations with the modeled daily concentrations derived uniformly from 00:00 UTC of the relevant day. We apologize for this oversight. We have recalculated the validation results for these two stations using the correct time windows and included the updated values in the relevant tables in the revised Supplement (Tables S10 and S12). We also revised the figures displaying the mean daily OC concentrations (Figs. S4 and S6 in the revised Supplement) and the differences between modeled and observed values at these two stations (Figs. 6 and 10 in the revised manuscript) to reflect these corrections. We made a similar mistake in the analysis of meteorological variables at these two stations. This has now been corrected to follow the procedure described in the revised manuscript, and the corresponding table (Table S8) and figures (Figs. 2 and S1) have been updated accordingly. Based on these updated analyses, we have made several corresponding revisions to the text describing the results to ensure consistency with the corrected values.

3. Some clarifications or explanations would be needed in the results section:

- L367-368: Could you please explain why you have chosen different emission estimates?

    We used different IVOC emission estimates because our approach is based on more recent smog chamber experiments specific to biomass burning sources (Jiang et al., 2021; Ciarelli et al., 2017), which suggest higher IVOC/POA ratios than the generic parameterization by Robinson et al. (2007). This clarification has been added to the revised manuscript.

- L430: What does the "similar conclusion" refer to here? Does it refer to the conclusion in the previous sentence about wind speed inaccuracy in the model being possibly the explanation for the underestimated OC? Why would that affect CVb and CVa most?

    In the original version, the phrase *"a similar conclusion"* was intended to refer to the fact that a similar pattern is observed — namely, that the Košetice station shows better prediction accuracy than the stations in the Kladensko and Třinecko areas during the summer seasons — and not to the explanation involving wind speed differences. We have clarified this in the revised manuscript by rephrasing the sentence as follows:

    *"During the summer seasons, a similar pattern is observed in that the Košetice station (NMSE = 49.2–291.0 %, IOA = 0.36–0.52, FAC2 = 0–56.7 %) shows better prediction accuracy than the stations in the Kladensko and Třinecko areas (NMSE = 114.2–653.8 %, IOA = 0.36–0.49, FAC2 = 0–31 %; Table S11), which perform even more poorly than during the winter phases."*

- Are the concentrations in the map figures (e.g. Fig. 4) surface level concentrations or, e.g., averaged through the vertical layers of the model?

These are the near-surface concentrations, or more precisely, concentrations in the first model layer, which spanned approximately 50 m in vertical extent. We have clarified this in the revised manuscript by stating it explicitly at the beginning of Sects. 3.2 and 3.3.

- L493-495: "The observed impacts in these simulations are likely linked to changes in other pollutant(s) at the boundaries of the model domain, which influence SOA chemistry." Could the authors please explain what these other pollutants are, how/why they changed and how that would affect the organic aerosol in the model?

We agree that further clarification was needed and have revised the text accordingly. In the updated manuscript, we now explicitly state that the impacts on SOA concentrations can be directly attributable to changes in the chemical composition of the chemical boundary conditions (CBCs), which were the only factor varied in the relevant simulations (i.e., Sp100s0 vs. CSwI and Vp100s0 vs. CVb). These changes may affect both the oxidative environment — through species such as ozone, nitrogen oxides, carbon monoxide and the hydroxyl radical — and the availability of direct SOA precursors, including toluene, xylene, and isoprene. This clarification has been added in the results section and supported by references to Sect. 2.3 in the revised manuscript and Tables S1–S2 in the revised Supplement, where the full CBC composition is documented.

- L495-496: "The spatial distributions of the mean seasonal impacts on SOA concentrations in Sp50s50 and Sp0s100 (and similarly in Vp50s50 and Vp0s100) exhibit structures akin to those observed for the mean seasonal impacts on POA concentrations in Sp50s50 and Sp100s0 (and likewise in Vp50s50 and Vp50s000) (Figs. 8c and d) during both seasons." Could the authors please comment if this is an expected result? Or does this point towards the boundary conditions defining too much the concentrations over the simulated area?

The observed resemblance can be partly understood by considering the volatility characteristics of the OA surrogate species at the model boundaries, as discussed in the revised manuscript. In both seasons, POA and SOA are predominantly redistributed into non-volatile and low-volatility surrogate species (Tables S4–S6 in the revised Supplement), which reside largely in the aerosol phase and are therefore efficiently transported. Given that all simulations are driven by the same meteorological conditions, this leads to similar transport behavior and helps explain the observed similarity in spatial patterns. This explanation has been included in the revised manuscript.

- L522-529: Why does the improvement with adding the OA at boundaries differ between the stations and seasons? Also, is it reasonable to assume that the OA at boundaries is only or mostly POA, i.e. do the authors expect that adding the OA as POA at boundaries is getting model results closer to the measured values because it is making the model representation of organic aerosols more accurate, or is the agreement better just because there is a large underestimation in the reference simulation and adding the OA at boundaries as POA happens to increase OA concentration most?

A new paragraph — now the penultimate paragraph in Sect. 3.3.2 of the revised manuscript — was added to address the first part of the comment. It explains that the improvement or deterioration in the modeled mean daily OC concentrations resulting from the addition of OA at the boundaries of the model domain differs between the stations and seasons analyzed due to the combined influence of several interacting factors that vary both spatially and temporally. These include (1) the annual variation in the mean monthly concentrations of the total OA prescribed at the boundaries of the model domain, (2) the seasonal variation in how these concentrations are redistributed into the POA and SOA surrogate species, (3) changes in atmospheric conditions that affect the transport and chemistry of OA (e.g., wind patterns and temperature), and (4) spatial and temporal variability in anthropogenic and biogenic emissions inside the model domain.

The final paragraph of Sect. 3.3.2 addresses the second part of the comment. It clarifies that the redistribution scenarios treating total OA at the boundaries of the model domain as entirely POA or entirely SOA were not intended to represent realistic conditions, but were

designed as bounding cases to assess the sensitivity of the model to the unknown OA composition in the EAC4 dataset by exploring the maximum plausible range of impacts on the modeled mean daily OC concentrations. Although Sp100s0 and Vp100s0 produced the mean daily OC concentrations that most closely matched the observations in this sensitivity study, they assumed OA to be entirely composed of POA at the boundaries of the model domain, which is highly unrealistic. As already noted in the Introduction, Chen et al. (2022) found that SOA dominates the organic aerosol fraction of $PM_1$ across Europe (ranging from 47.3 % to 100 %), indicating that a significant SOA component may be expected in real boundary conditions. While their results pertain to $PM_1$, they suggest that the improvements in the modeled mean daily OC concentrations obtained in Sp50s50 and Vp50s50, or in simulations falling between Sp50s50 and Sp0s100 and between Vp50s50 and Vp0s100, may more realistically reflect the influence of OA composition at the boundaries of the model domain.

4. This study includes sensitivity analysis on estimates of IVOC and SVOC emissions and OA boundary conditions, as well as comparisons using two different SOA schemes and chemistry schemes. Is it possible to conclude which of the analyzed factors/assumptions, or their uncertainties, are most important from the point of view of modelling OA in Central Europe with this model?

The revised conclusions now address the relative importance of the analyzed factors. They clarify that the setup used in the CVa experiment—featuring the 1.5-D VBS scheme with aging from all OA sources and the inclusion of both source-specific and non-source-specific IVOC and SVOC emissions—resulted in the best overall model performance, particularly during winter. They also explain that the inclusion and composition of OA in the boundary conditions had the greatest impact during summer, especially at rural sites. While the two sensitivity analyses targeted different aspects and cannot be directly compared, the conclusions now highlight how the dominant source of model sensitivity varies seasonally, providing clearer guidance on which assumptions are most influential for OA modeling in Central Europe.

**Minor comments**

Please mention in the abstract that the evaluation of the model simulations is focused on Czech Republic. Currently the reader finds out quite late in the text that the evaluation is not for wider Central Europe but only for one country.

We have added this information to the abstract.

L21: "have an undoubted environmental footprint" Please check the choice of word. In my understanding environmental footprint term is used for the impact of e.g. organization or products on environment, so for the source of aerosols one could talk about environmental footprint, but not for aerosols themselves. I did not find the term "environmental footprint" from the reference given for this statement, therefore it is not clear what the authors mean by this term.

We agree that *"environmental footprint"* is not the appropriate term in this context. To better reflect our intended meaning—that aerosols represent a burden to the environment—we have revised the wording to *"environmental burden."*

L136-137: I would suggest moving this essential information from the Appendix to the main text.

In the revised manuscript, we have followed this suggestion and moved the content previously presented in Appendix A into the main text. This material now appears as a dedicated subsection (Sect. 2.2), ensuring that the description and comparison of the SOAP and 1.5-D VBS schemes are more prominently integrated into the Methods section. We believe this change improves both clarity and accessibility for the reader.

Some of the figures, e.g. Figure 2, are missing y-axis labels. I recommend adding y-axis labels.

Y-axis labels have now been added to all relevant figures (Figs. 2, 3, 6, 7, 10, 11 in the revised manuscript, and Figs. S1–S7 in the revised Supplement). Additionally, we have added descriptive titles to these figures to clarify the meaning of the symbols used in the y-axis labels. We hope these changes enhance the clarity and readability of the figures.

Tables S3, S4 and S5 contain acronyms for surrogate SOA/POA species in the model. Please add explanation of what these species are.

We have added footnotes to the relevant tables (Tables S4–S6 in the revised Supplement), which provide explanations of the surrogate SOA/POA species used in the model.
* * *
References:

Ciarelli, G., Aksoyoglu, S., El Haddad, I., Bruns, E. A., Crippa, M., Poulain, L., Äijälä, M., Carbone, S., Freney, E., O'Dowd, C., Baltensperger, U., and Prévôt, A. S. H.: Modelling winter organic aerosol at the European scale with CAMx: evaluation and source apportionment with a VBS parameterization based on novel wood burning smog chamber experiments, Atmospheric Chemistry and Physics, 17, 7653–7669, https://doi.org/10.5194/acp-17-7653-2017, 2017.

Huszar, P., Karlický, J., Ďoubalová, J., Šindelářová, K., Nováková, T., Belda, M., Halenka, T., Žák, M., and Pišoft, P.: Urban canopy meteorological forcing and its impact on ozone and PM2.5: role of vertical turbulent transport, Atmospheric Chemistry and Physics, 20, 1977–2016, https://doi.org/10.5194/acp-20-1977-2020, 2020.

Jiang, J., El Haddad, I., Aksoyoglu, S., Stefenelli, G., Bertrand, A., Marchand, N., Canonaco, F., Petit, J.-E., Favez, O., Gilardoni, S., Baltensperger, U., and Prévôt, A. S. H.: Influence of biomass burning vapor wall loss correction on modeling organic aerosols in Europe by CAMx v6.50, Geoscientific Model Development, 14, 1681–1697, https://doi.org/10.5194/gmd-14-1681-2021, 2021.

Ramboll: CAMx User's Guide, Comprehensive Air Quality model with Extentions, version 7.10, Novato, California, https://www.camx.com/download/source/, (last access: 5 June 2025), 2020.

Robinson, A. L., Donahue, N. M., Shrivastava, M. K., Weitkamp, E. A., Sage, A. M., Grieshop, A. P., Lane, T. E., Pierce, J. R., and Pandis, S. N.: Rethinking Organic Aerosols: Semivolatile Emissions and Photochemical Aging, Science, 315, 1259–1262, https://doi.org/10.1126/science.1133061, 2007.

---

## Author Comment (AC2)

**Authors' response to Anonymous Referee #2**

**on review of** *"Modeling organic aerosol over Central Europe: uncertainties linked to different chemical mechanisms, parameterizations, and boundary conditions"*

**by Lukáš Bartík et al. (ecusphere-2025-167 )**

Dear Anonymous Referee #2,

We sincerely thank you for the time and effort you dedicated to reviewing our manuscript, and for your constructive and insightful comments. Please find below our detailed, point-by-point responses (in black) to the comments you provided (in blue).

The study presents some key challenges in accurately modeling organic aerosol concentrations over Central Europe. By using an advanced regional model along with the different approaches for atmospheric aging of organic matter, it identifies the importance of including explicit SVOC and IVOC parameterizations in emission inventories to achieve higher agreement with observations. Moreover, it points out that the share between primary and secondary organic aerosol considered for the boundary conditions is a critical choice for accuracy. This study explores a region that has not been the main focus of air quality research in Europe and would therefore be a useful reference for the community. Before I can suggest it for final publication, there are some major and minor issues that should be first addressed. You can find them below :

**Major comments:**

- What I found missing was a comprehensive round-up in Section 4 regarding the rate of improvement among the different model setups that were tested. For example, it is mentioned in Section 3 that the first sensitivity experiment provided increased improvement for the winter period, but the second one did so for the summer period. However different location types were taken into account as well (Rural/Urban), and it was not made clear whether one particular setup is more suitable depending on those conditions. The authors should expand the conclusions in order to clearly present which model configuration is the most important option depending on location type and season.

  The conclusions have been revised to more clearly summarize the relative performance of the tested model setups by season and station type, as requested. They now indicate that the CVa configuration—combining the 1.5-D VBS scheme with aging from all OA sources and the inclusion of IVOC/SVOC emissions—provided the best overall model performance, particularly during winter. It is also noted that, under this configuration, rural stations were generally better predicted than urban stations in both seasons. The revised text further explains that adding OA to the boundary conditions had the strongest influence during summer, especially at rural sites. Together, these points clarify which configurations are most suitable depending on season and location.

**Minor comments:**

- Line 12: "the accuracy of modeled organic carbon concentrations improving by up to 100 %". This sentence should be rephrased as the improvement corresponds to the FAC2 metric specifically, and can otherwise be misleading.

  To avoid potential misinterpretation, we have removed the quantitative information and now refer only to *"significant improvements"* during summer at some monitoring sites.

- Line 47: The authors should consider presenting the saturation concentrations of LVOCs and SVOCs in a similar format as those for the IVOCs.

  We have standardized the numerical format of all effective saturation concentrations to match the format used for IVOCs in the original manuscript.

- Line 55: " both the original 1.5-D VBS and its various modifications". What do these modifications alter in the original 1.5-D VBS framework?

  In general, these modifications differ from the original 1.5-D VBS scheme (Koo et al., 2014) in the number of basis sets used and/or in the physical parameters that characterize them. For example, the implementation presented by Woody et al. (2016), used in our CAMx simulations, introduced five basis sets instead of four, while other studies, such as Jiang et al. (2019), further divided these into more source-specific sets and updated key parameters based on smog chamber data. These changes allow for a more detailed and source-resolved representation of OA formation and aging. We have briefly clarified this point in the revised manuscript by specifying that the modifications involve changes to the number of basis sets, the physical parameters that define them, or both.

- Line 121: What was the reasoning behind using 2 different chemical mechanisms in your simulations, since one is more comprehensive than the other?

  Our motivation for this investigation was fundamentally practical. Given that both families of gas-phase mechanisms—the Statewide Air Pollution Research Center (SAPRC) mechanisms and Carbon Bond (CB) mechanisms—are widely used in the modeling community, we aimed to examine how differences in gas-phase chemistry formulations affect secondary organic aerosol (SOA) production, while keeping the SOA treatment unchanged.

  To clarify this, we have added a new short paragraph in Sect. 2.1 of the revised manuscript explaining our rationale for using both mechanisms. Additionally, we note in this paragraph that their use also reflects practical constraints in the available configurations of the CAMx model: only certain combinations of gas-phase mechanisms and OA modules are supported without requiring modifications to the model code, which we sought to avoid.

- Lines 127–128: Is the aerosol size distribution in the CAMx model bimodal too? Are there any kinetic limitations taken into account for gas diffusion?

  Yes, CAMx considers fine  and coarse modes (0–2.5 µm and 2.5–10  µm, respectively). The coarse fraction is treated as completely inert, so organic aerosol (both primary and secondary) as well as secondary inorganic aerosol are expected to reside within the fine fraction. Moreover, all aqueous-phase chemistry occurs within the fine mode. Gas diffusion within the water condensed on aerosols is not explicitly considered; it is assumed that the gas concentration becomes uniform within the aqueous phase after absorption. Due to the sufficiently large size of aerosol particles, and thus the large corresponding aqueous spheres, kinematic effects for gas transport to the surface are also neglected.

  We would also like to mention that, based on the recommendation of Referee #1, we have expanded the penultimate paragraph of Sect. 2.1 in the revised manuscript to provide a clearer explanation of how aerosols are represented in the model simulations.

- Lines 150–152: Does the REZZO inventory have a different spatiotemporal resolution than CAMS? What do you mean by 'temporal disaggregation and speciation'?

Yes, while the CAMS inventory has spatial resolution of 0.05° x 0.1° corresponding to an area of 5 km x 10 km, the REZZO emission data offers detailed information on individual point sources as well as emission from so called *"basic territorial units"* which often corresponds to small areas of a few 100 m x 100 m (up to a few 1 km). Transport emissions from the ATEM dataset are also defined for individual roads (and interpolated to the model grid using FUME) so have also higher resolution than CAMS. We have made this clear in the revised text.

Temporal disaggregation means how the annual emissions are decomposed into hourly emission data. This is done using temporal factors describing the evolution of emissions from different activity sectors throughout the year, during the week and during the day. Speciation refers to the decomposition of total non-methane VOC emissions into individual gas-phase compounds (such as ethane, ethene, acetaldehyde, etc.) or families of compounds, such as higher aldehydes and higher ketones. It also includes the decomposition of emitted fine particulate matter into black carbon, primary organic matter, and other fine inert material. We have added this information in the revised text.

- Section 2.3: I recommend that the authors change the titles of subsections 2.3.1 & 2.3.2 in order to make clear what was changed in each sensitivity analysis.

We have extended these titles to be more specific, explicitly mentioning what was changed in each of the sensitivity analyses.

- Table 1: Why was the VBS framework not combined with the SAPRC07TC chemical mechanism at all for the CVb and CVa experiments?

Our approach was to establish a reference case using the SOAP mechanism and CB6r5 gas-phase chemistry without IVOC and SVOC emissions, and then systematically examine the sensitivity to changes in these settings—namely, by adding I/SVOC emissions, altering the gas-phase chemical mechanism, or switching the OA module.

As we explain in the newly inserted paragraph in Sect. 2.1 of the revised manuscript, when designing the experiments described in Sect. 2.4, we considered only those combinations of gas-phase mechanisms and OA modules that are directly supported by the CAMx model. Other pairings would have required additional modifications to the model code, which we sought to avoid.

Since the combination of SAPRC07TC with the 1.5-D VBS framework is not implemented in the CAMx version we used, it was not included in our experimental setup. We have clarified this rationale in the revised manuscript, and we hope this explanation sufficiently addresses the referee's concern.

- Lines 223 & 224: Some of the factors for SOA redistribution in Tables S3 & S4 are higher in the winter than in the summer. Since typically SOA formation peaks in the warmer periods, is this something you expect?

We agree with the referee that SOA formation typically peaks during warmer periods, and we appreciate this insightful observation. However, the values presented in Tables S4 and S5 in the revised Supplement (formerly, Tables S3 and S4) are seasonal redistribution factors derived as normalized fractions — calculated independently for each season and for each reference experiment. Specifically, for each season, we divided the domain-averaged mean concentration of each surrogate SOA species by the seasonal total SOA concentration (i.e., the sum of all SOA surrogate species in that experiment and season).

These values thus represent the relative composition of SOA among its surrogate species for a given season, not the absolute concentration of SOA. Therefore, comparing values across rows (e.g., winter vs. summer) does not reflect seasonal variations in total SOA formation, but rather how SOA is distributed among the individual surrogate species within each season.

We acknowledge that the previous version of the manuscript may have left some room for ambiguity in this regard. To address this, we have revised the relevant paragraph (see Sect. 2.4.2 in the revised manuscript) to clarify both the purpose and normalized nature of these factors.

- Lines 224 & 225: Similarly, some of the factors for POA redistribution in Table S5 are higher in the summer than in the winter. Since, typically during the colder period POA emissions peak due to combustion generated for heating demands, is this something you expect?

We thank the referee for this comment, which parallels the previous one on SOA. The values in Table S6 of the revised Supplement (formerly Table S5) are likewise seasonal redistribution factors derived as normalized fractions. They represent the relative composition of POA among its surrogate species in each season and do not reflect the absolute amounts of those species. We have clarified this point in the revised manuscript (see Sect. 2.4.2).

- Section 2.4: What is the frequency of the model output regarding aerosol concentrations? Does it produce average (i.e. daily) or instantaneous values?

CAMx offers hourly averaged outputs for all species, both gas-phase and aerosol (i.e. these are mean concentrations over each hour, not instantaneous values).

Here, we would also like to inform the referee that, in addition to the changes discussed with Referee #1 regarding Sect. 2.5, 'Validation', in the revised manuscript (formerly Sect. 2.4), we revised the final two paragraphs of this section to improve clarity and precision. In the penultimate paragraph of the original version, we incorrectly implied that meteorological variables were not measured at the Prague–Suchdol station. In the revised manuscript, we clarify that Prague–Suchdol is an air quality station where accompanying meteorological measurements are indeed conducted. However, the relevant meteorological data from this site were not included in the dataset provided to us, which was limited to professional meteorological stations. For this reason, we used data from the nearby Prague–Kbely station—a professional station—which we selected as a representative site for Prague–Suchdol.

In the final paragraph, we clarified the rationale for comparing daily OC concentrations—namely, to ensure at least consistency in the duration of sampling periods across all the stations, as the longest period among them was 24 hours (at the Prague–Suchdol station), even though the sampling windows differ between sites. We also specified that CAMx was configured to output hourly averaged concentrations, which allowed us to construct daily model outputs that matched the sampling periods at each station. Additionally, we made clear that for the meteorological evaluation, only the mean daily modeled and observed values from the days with available OC measurements were used, with each daily value constructed to follow the station-specific 24-hour sampling windows.

As such, the answer to your question is now explicitly stated in the revised manuscript.

Importantly, we would like to point out that in the case of the analysis for the Prague–Suchdol and Košetice stations, we made a mistake by comparing the measured daily OC concentrations with the modeled daily concentrations derived uniformly from 00:00 UTC of the relevant day. We apologize for this oversight. We have recalculated the validation results for these two stations using the correct time windows and included the updated values in the

relevant tables in the revised Supplement (Tables S10 and S12). We also revised the figures displaying the mean daily OC concentrations (Figs. S4 and S6 in the revised Supplement) and the differences between modeled and observed values at these two stations (Figs. 6 and 10 in the revised manuscript) to reflect these corrections. We made a similar mistake in the analysis of meteorological variables at these two stations. This has now been corrected to follow the procedure described in the revised manuscript, and the corresponding table (Table S8) and figures (Figs. 2 and S1) have been updated accordingly. Based on these updated analyses, we have made several corresponding revisions to the text describing the results to ensure consistency with the corrected values.

- Lines 245 & 246: Were daily values extrapolated for the validation in this case, or was only the model output corresponding to the measurement dates used?

See our response above.

- Line 279: "the model typically tends to overestimate them more or less". It would look better if a more accurate measure for the overestimation was stated here.

We rephrased this sentence to remove ambiguous statements and it now simply writes: *"Regarding the mean daily wind speeds, the model has the tendency to overestimate them …"*

- Figures 2 & 3: The authors should consider adding a legend with information about what a positive/negative difference corresponds to, similarly to how Figures 6 & 7 are presented.

We have added y-axis labels to all relevant figures (Figs. 2, 3, 6, 7, 10, 11 in the revised manuscript, and Figs. S1–S7 in the revised Supplement). Additionally, we have added descriptive titles to these figures to clarify the meaning of the symbols used in the y-axis labels. We hope these changes enhance the clarity and readability of the figures.

- Lines 291–292: The underestimation of both temperature and relative humidity over the domain, also plays a role in that result. It would be important to point that out.

We agree that the underestimation of both temperature and relative humidity may influence the modeled aerosol concentrations. To address this point, we have added the following as the final paragraph of Sect. 3.1 in the revised manuscript:

*"Finally, it is also worth noting that the model biases in the other studied meteorological conditions may influence the modeled aerosol concentrations. For example, the lower modeled temperatures lead to an underestimation of gas-phase reaction rates due to their temperature dependence, but they may also enhance gas-to-particle partitioning. The negative bias in the modeled relative humidity compared to the observations affects particle size and density, as both are influenced by the aerosol water content determined by the local humidity.…"*

- Line 328: Does that mean that the concentrations in the CVb & CVa experiments are double those of the CSnI experiment?

To clarify this point, we have revised the paragraph in the manuscript to explicitly include the relative increases in mean seasonal POA concentrations in the CVb and CVa experiments compared to CSnI. Specifically, we now indicate that, compared to CSnI, the mean seasonal POA concentrations in CVb are higher on average by a factor of 1.65 and 1.74 in the winter and summer seasons, respectively, and in CVa by a factor of 1.66 and 1.80. These values are close to a doubling of concentrations, although still somewhat lower on average.

- Line 329: What do these "similarities" refer to?

In the original version of the manuscript, the sentence *"These similarities result from the scaling of SVOC emissions using both POA for most anthropogenic sources and NMVOC emissions for diesel and gasoline vehicles…"* was intended to refer specifically to the spatial similarities between the distributions of the mean seasonal impacts in CVb and CVa and the mean seasonal POA concentrations in CSnI. That is, the observed resemblance in spatial patterns results from the way SVOC emissions were scaled.

To improve clarity, we have rephrased the sentence to make the intended reference more explicit and to better integrate it into the discussion of results. The revised sentence now reads: *"This resemblance in spatial patterns can be explained by the way SVOC emissions were scaled: POA emissions were used for most anthropogenic sources, while NMVOC emissions were used for diesel and gasoline vehicles, whose spatial distributions closely match those of POA emissions from the same vehicle categories."*

- Figure 5: The relative increase in SOA concentrations by the CVb experiment is much more drastic (1 order of magnitude compared to CSnI) in winter than in summer, around the Po Valley. How can this be explained?

The more pronounced increase in SOA concentrations in CVb during winter (compared to CSnI), particularly in the Po Valley, is due to higher wintertime emissions of IVOCs and SVOCs from residential biomass burning. These emissions are significantly reduced in summer, leading to a smaller SOA response.

We have added this explanation to the third paragraph of Sect. 3.2.1, directly after the discussion of the CVb results, with the sentence: *"The relatively smaller summer increase, compared to the winter increase, over regions such as the Po Valley, the Czech Republic, and the Pannonian Basin can be attributed to the seasonal reduction in IVOC and SVOC emissions from residential biomass burning."*

- Line 361: Please change 'he' to 'they' when referring to a study.

Changed.

- Lines 362–365: If CSnI is taken as the reference experiment in this sensitivity, why is the comparison here with the results of Meroni et al., (2017) made with the CSwI experiment?

Although CSnI is the reference experiment for the sensitivity analysis, the comparison with the experiment by Meroni et al. (2017) was made using the CSwI experiment, as their model configuration is more consistent with that of CSwI. Specifically, they used SOAP to represent OA chemistry, the CB05 mechanism (Yarwood et al., 2005) for gas-phase chemistry, and included IVOC emissions. In particular, the inclusion of IVOC emissions is a key feature of CSwI but is not present in CSnI. Therefore, CSwI provides a more appropriate basis for comparison. We have revised the corresponding paragraph in the manuscript to clarify this rationale more explicitly.

- Lines 375 & 376: In Table 1 it states that this particular experiment did include SVOC emissions. Please clarify.

We acknowledge that the original sentence did not clearly convey our intended message and may have caused confusion regarding the inclusion of SVOC emissions. Specifically, we intended to emphasize that our assumption about the absence of SVOC emissions in the

emission inventories used to prepare input emission data for our experiments (REZZO, CAMS-REG-v4.2, and ATEM; see Sect. 2.3) is only partially accurate. We have revised the sentence accordingly to clarify this point. To further address your comment, we confirm that SVOC emissions were included in the CVa experiment, as indicated in Table1.

- Section 3.2.2: For the 2nd paragraph please provide percentages as a measure for the differences. The authors should also consider changing the order of paragraphs 3 & 4 to match the order of presented results by figures and tables.

In the revised manuscript, we have expanded and clarified the former second paragraph by including mean percentage differences (MPDs) between the modeled and observed daily OC concentrations, thereby providing a clearer quantitative comparison. We have also reordered the paragraphs as suggested to align with the order of figures and tables and have slightly modified them accordingly.

- Line 416: Shouldn't Tables S9 and S10 be referenced here, instead of S4 and S6?

You are correct — this was a mistake. Tables S9 and S10 from the original Supplement should have been referenced there. In the revised manuscript, we have corrected the reference accordingly. It now points to Tables S10 and S11 in the revised Supplement (formerly Tables S9 and S10 in the original Supplement).

- Lines 474–478: Is this behavior driven solely by the fractions of PAP0 and PAP1? And if so, is it expected? Are the impacts of PFP0,PFP1 and PFP2 not as important?

Upon review, we found that the original statement lacked a clear reference to which simulations were being compared, and we apologize for this oversight. In the revised manuscript, we have reformulated the comparison to explicitly state that the increase in POA impacts is evaluated relative to Sp0s100 and Vp0s100, depending on the OA scheme used.

We have also expanded the explanation to clarify that the observed differences are not solely driven by PAP0 and PAP1, but rather by the redistribution of POA at the boundaries of the model domain among the POA surrogate species with different volatilities (Table S6), and by the treatment of volatility in the OA schemes. Specifically, in the VBS-driven simulations (Vp50s50 and Vp100s0), only a portion of the POA (PAP0 and PFP0) is treated as purely non-volatile. The remaining portion of this added POA can partly evaporate inside the model domain, and part of the evaporated material can be further aged to form SOA. In contrast, in the SOAP-driven simulations (Sp50s50 and Sp100s0), the same total POA as in Vp50s50 and Vp100s0, respectively, enters the model domain from the boundaries, but it is treated entirely as non-volatile, preventing both evaporation and subsequent aging.

- Line 494: "likely linked to changes in other pollutant(s)". Could you provide some examples?

We agree that further clarification was needed and have revised the text accordingly. In the updated manuscript, we now explicitly state that the impacts on SOA concentrations can be directly attributable to changes in the chemical composition of the chemical boundary conditions (CBCs), which were the only factor varied in the relevant simulations (i.e., Sp100s0 vs. CSwI and Vp100s0 vs. CVb). These changes may affect both the oxidative environment — through species such as ozone, nitrogen oxides, carbon monoxide, and the hydroxyl radical — and the availability of direct SOA precursors, including toluene, xylene, and isoprene. This clarification has been added in the results section and supported by references to Sect. 2.3 in the revised manuscript and Tables S1–S2 in the revised Supplement, where the full CBC composition is documented.

- Line 497: Shouldn't it be 'Sp0s100' instead of 'Sp100s0' ? And similarly, shouldn't it be 'Vp0s100' instead of 'Vp50s000'?

The reference to Sp100s0 is correct, as it was used intentionally to compare the impact on SOA in Sp0s100 with the impact on POA in Sp100s0. However, the second case involved a typographical error — Vp50s000 should indeed have been Vp100s0. We have corrected this in the revised manuscript and also split the sentence into two for clarity.

- Lines 500 & 501: Why would the evaporation of POA be relevant for the p0s100 scenarios?

In response to this comment, we see no reason for the evaporation of POA to be relevant in the p0s100 scenarios. However, upon closer inspection of the results, we found that the statement preceding the reasoning discussed here — *"The patterns of these distributions, along with the values of ΔSOA, indicate that the increase in the mean seasonal impacts as the proportion of SOA in OA at boundaries increases is consistently higher during both seasons in experiments where SOAP handles OA chemistry."* — is valid only in terms of domain-averaged values. The spatial distributions reveal more complexity, and the statement does not hold generally across the domain. Therefore, we have removed both the original statement and its associated reasoning from the original manuscript and did not retain them in the revised version.

- Lines 501–505: Is this behavior driven solely by the fractions of PAS0/PBS0 (VBS module) and SOA4/SOPB (SOAP module)? Are the impacts of others not as important?

Upon closer inspection of the results, we found that the original statements in the part of the paragraph referenced by the referee — describing the seasonal contrasts in the mean seasonal impacts on SOA concentrations in Sp50s50, Vp50s50, Sp0s100, and Vp0s100 — did not fully capture the spatial variability across the domain. Although the domain-averaged values of these impacts are higher during the summers in all four simulations, the mean seasonal impacts are not consistently higher across the entire domain. In some areas localized within the southeastern part of the domain, the impacts are actually higher during the winters — by up to $0.1\,\mu g\,m^{-3}$ in Sp50s50 and Vp50s50, and by up to $0.5\,\mu g\,m^{-3}$ in Sp0s100 and Vp0s100. To address this, we have revised the relevant text in the original manuscript, introducing a new paragraph in the revised version to more accurately reflect this spatial complexity.

Regarding the origin of this behavior, we clarify that it is not solely driven by the fractions of PAS0/PBS0 (in the VBS module) or SOA4/SOPB (in the SOAP module). Rather, it is driven by a combination of the seasonal variation in the mean monthly OA concentrations at the boundaries, which affects all four simulations, and the redistribution of these concentrations among (1) the SOA surrogate species in Sp0s100 and Vp0s100, and (2) both the POA and SOA surrogate species in Sp50s50 and Vp50s50. This clarification has also been added to the new paragraph in the revised manuscript.

- Table S2: What is the size distribution of organic matter, black carbon and sulfate? Did it have to change for the mapping? Where there any CBCs considered for ammonium, nitrate and mineral cations besides sodium, since they are also treated by ISORROPIA? Was dust considered in a bulk only state without any particular chemical composition?

All organic matter present in the EAC4 data was considered as fine aerosol, so it was entirely included in CAMx's fine aerosol bin (where POA/SOA belong). Sulphates were also considered as fine aerosol only; the same applies to black carbon. This means that we did not need to consider any modifications in size distributions. Ammonium and nitrates were not

included in this reanalysis. Mineral cations were explicitly mapped only for sea salt, which was treated as a mixture of sodium, chloride, sulphate, and magnesium. Dust was therefore taken into account as bulk matter without chemical speciation.

- Table S3: How are the 6 SOA surrogate species allocated? In Appendix A it is mentioned that SOAP considers 3 species (2+1) from anthropogenic and another 3 (2+1) from biogenic. Can you specify which is used for what?

We have added a footnote to the relevant table (Table S4 in the revised Supplement), which provides an explanation of the six surrogate SOA species used in SOAP.

- Tables S3, S4 and S5: The authors should consider inserting footnotes to explain what all the species abbreviations refer to.

We have added footnotes to the relevant tables (Tables S4–S6 in the revised Supplement), which provide explanations for all the surrogate SOA and POA species abbreviations used.

- Table S6: If I understood correctly, the measurements from the Prague-Schudol station correspond to $PM_{10}$. This should be clarified in the footnotes.

We have added a new footnote to Table S7 in the revised Supplement (formerly Table S6), which clearly states that measurements at the Prague–Suchdol station represent OC within the $PM_{10}$ fraction, while measurements at all other stations represent OC within the $PM_{2.5}$ fraction.

- Table S9: What do the last 3 columns correspond to?

Table S10 in the revised Supplement (formerly Table S9) presents statistical analysis of the mean daily OC concentrations predicted by the individual experiments of the first sensitivity analysis at the Prague–Suchdol and Košetice stations for the individual seasons. The last three columns present analogous statistical analysis—using the same set of statistical metrics (MB, RMSE, NMSE, IOA, FAC2)—for the Košetice station, but evaluated over time periods corresponding to the winter campaign phases in the Třinecko (Tř-w) and Kladensko (Kl-w) areas, and the summer campaign phase in the Kladensko (Kl-s) area, respectively.

To improve clarity, we have revised the caption of Table S10 in the Supplement to explicitly describe the meaning of these columns. As a result, we have also removed the three corresponding footnotes, which are now redundant.

- Figure S3: I believe that this figure is redundant and can be omitted, as this is more general information that can be easily accessed in statistics books.

We agree that this information is generally available in statistics textbooks and have therefore decided to remove the figure from the Supplement.

References:

Meroni, A., Pirovano, G., Gilardoni, S., Lonati, G., Colombi, C., Gianelle, V., Paglione, M., Poluzzi, V., Riva, G., and Toppetti, A.: Investigating the role of chemical and physical processes on organic aerosol modelling with CAMx in the Po Valley during a winter episode, Atmospheric Environment, 171, 126–142, https://doi.org/10.1016/j.atmosenv.2017.10.004 , 2017.
* * *
References:

Jiang, J., Aksoyoglu, S., El-Haddad, I., Ciarelli, G., Denier van der Gon, H. A. C., Canonaco, F., Gilardoni, S., Paglione, M., Minguillón, M. C., Favez, O., Zhang, Y., Marchand, N., Hao, L., Virtanen, A., Florou, K., O'Dowd, C., Ovadnevaite, J., Baltensperger, U., and Prévôt, A. S. H.: Sources of organic aerosols in Europe: a modeling study using CAMx with modified volatility basis set scheme, Atmos. Chem. Phys., 19, 15247–15270, https://doi.org/10.5194/acp-19-15247-2019, 2019.

Koo, B., Knipping, E., and Yarwood, G.: 1.5-Dimensional volatility basis set approach for modeling organic aerosol in CAMx and CMAQ, Atmospheric Environment, 95, 158–164, https://doi.org/10.1016/j.atmosenv.2014.06.031, 2014.

Meroni, A., Pirovano, G., Gilardoni, S., Lonati, G., Colombi, C., Gianelle, V., Paglione, M., Poluzzi, V., Riva, G., and Toppetti, A.: Investigating the role of chemical and physical processes on organic aerosol modelling with CAMx in the Po Valley during a winter episode, Atmospheric Environment, 171, 126–142, https://doi.org/10.1016/j.atmosenv.2017.10.004, 2017.

Woody, M. C., Baker, K. R., Hayes, P. L., Jimenez, J. L., Koo, B., and Pye, H. O. T.: Understanding sources of organic aerosol during CalNex-2010 using the CMAQ-VBS, Atmospheric Chemistry and Physics, 16, 4081–4100, https://doi.org/10.5194/acp-16-4081-2016.

Yarwood, G., Rao, S., Yocke, M., and Whitten, G. Z.: Updates to the Carbon Bond Chemical Mechanism: CB05, Final Report RT-04-00675 prepared for US EPA, https://www.camx.com/Files/CB05_Final_Report_120805.pdf, 2005.

---

## Author Response (AR2)

**Authors' response to Anonymous Referee #1**

**on review of** "Modeling organic aerosol over Central Europe: uncertainties linked to different chemical mechanisms, parameterizations, and boundary conditions"

by Lukáš Bartík et al. (ecusphere-2025-167)

Dear Anonymous Referee #1,

We sincerely thank you for the time and effort you dedicated to reviewing our manuscript, and for your constructive and insightful comments. Please find below our detailed, point-by-point responses (in black) to the comments you provided (in blue).

The authors have addressed my comments and revised the manuscript sufficiently. I can recommend publication of the manuscript, but I have one further suggestion. In response to one of my comments, the authors have added the paragraph L364-639 discussing the CBCs and their effect. I find that this paragraph would benefit from further clarification of explicitly stating that the gas phase CBCs were also different in these sensitivity simulations compared to the basecase simulations CSwI and CVb.

The paragraph has been revised to clarify that the CBCs differed for both gas-phase and aerosol species in the sensitivity simulations compared to the reference simulations. The revised text (lines 637–642) now reads:

"Since the CBCs were the only factor varied between the simulations used to quantify these impacts (i.e., Sp100s0 vs. CSwl and Vp100s0 vs. CVb), the resulting differences in SOA concentrations can be directly attributed to modifications to the CBCs for both gas-phase and aerosol species. These modifications may affect both the oxidative environment, through species such as ozone, nitrogen oxides, carbon monoxide, and the hydroxyl radical, and the availability of direct SOA precursors such as toluene, xylene, and isoprene, helping to explain the spatial and seasonal variation observed. As mentioned earlier, the detailed composition of the two CBC sets is provided in Sect. \ref{inputs} and Tables S1--S2."

It would be also helpful for the reader if this would be made more clear both when describing the sensitivity study and when starting to discuss these results (e.g. at the beginning of section 3.3 where the text is currently not stating what kind of sensitivity analyses these are). As it is currently described and discussed, the reader may easily get the wrong impression that the only difference here is the boundary condition for the aerosol phase. If there were also simulations with just the different gas phase CBC compared to the basecase, i.e. "Vp0s0" and "Sp0s0", it might be more clear. But as such simulations are missing, bit more clarification would be helpful.

Clarifications have been added in two locations in the manuscript to address this comment: in Sect. 2.4.2, where the sensitivity simulations are described, and at the beginning of Sect. 3.3, where the corresponding results are introduced. These revisions specify that both gas-phase and aerosol-phase CBCs differ from those in the reference simulations.

**Section 2.4.2:**

The text describing the second sensitivity analysis was revised to make clear that the chemical boundary conditions (CBCs) differed from those in the reference experiments not only for aerosol species but also for gas-phase species.

Specifically, the sentence:

"Each of these sensitivity experiments was performed using the same model setup and IVOC and  $POM_{SV}$  parameterizations as in its corresponding reference experiment, except for the chemical boundary conditions."

was replaced by:

"Each of these sensitivity experiments was performed using the same model setup and IVOC and  $POM_{SV}$  parameterizations as its corresponding reference experiment, but with CBCs that differed from those prescribed in the reference experiments (i.e., the default CBCs) in both gas-phase and aerosol species."

Additionally, in the next paragraph, the phrase:

"We then added the same boundary conditions for the remaining remapped aerosol species to each pair of these boundary conditions..."

was revised to:

"We then added the same EAC4-derived boundary conditions for all gas-phase species and for the remaining remapped aerosol species to each pair of these boundary conditions..."

**Section 3.3:**

The introductory paragraph of Sect. 3.3 was rewritten to clarify the nature of the sensitivity simulations and to explicitly state that both gas-phase and aerosol-phase CBCs were modified relative to the reference simulations.

The former text read:

"To present and discuss the results of this sensitivity analysis, we adopt a similar approach to that employed for the previous sensitivity study. Thus, we first investigate the spatial distributions of the mean seasonal impacts on the near-surface concentrations of POA and SOA in the experiments of this sensitivity analysis during both seasons, using the same definition of these impacts as in the first sensitivity study. Subsequently, we evaluate the OC concentrations obtained from the individual experiments of this sensitivity analysis."

The paragraph was revised as follows:

"This section presents the results of the second sensitivity analysis, in which the sensitivity experiments employed CBCs that differed from those prescribed in the reference experiments by modified gas-phase and additional aerosol species, as described in Sect. 2.4.2. To present and discuss these results, we follow a similar approach to that used in the previous sensitivity study. We first examine the spatial distributions of the mean seasonal impacts on the near-surface concentrations of POA and SOA in the experiments of this sensitivity analysis during both seasons, applying the same definition of these impacts as in the first sensitivity study. Finally, we evaluate the OC concentrations obtained from the individual experiments of this sensitivity analysis."

**Authors' response to Anonymous Referee #2**

**on review of** "Modeling organic aerosol over Central Europe: uncertainties linked to different chemical mechanisms, parameterizations, and boundary conditions"

by Lukáš Bartík et al. (ecusphere-2025-167)

Dear Anonymous Referee #2,

We sincerely thank you for your time and positive assessment recommending acceptance of our manuscript.